# TAVAE: A VAE with Adaptable Priors Explains Contextual Modulation in the Visual Cortex

**Balázs Meszéna**[1,*]**, Keith T. Murray**[1,2,*]**, Julien Corbo**[3]**, O. Batuhan Erkat**[3]**, Márton A. Hajnal**[1]**, Pierre-Olivier Polack**[3] **& Gergő Orbán**[1]

1, Department of Computational Sciences HUN-REN Wigner Research Centre for Physics
Budapest, Hungary
2, Princeton Neuroscience Institute, Princeton University, Princeton, NJ, US
3, Center for Molecular and Behavioral Neuroscience, Rutgers University, Newark, NJ, US
{meszena.balazs,hajnal.marton,orban.gergo} @wigner.hun-ren.hu,
km3199@princeton.edu,
julien.corbo@gmail.com,
oerkat@sunyopt.edu,
polack.po@rutgers.edu

## Abstract

The brain interprets visual information through learned regularities, a computation formalized as performing probabilistic inference under a prior. The visual cortex establishes priors for this inference, some of which are delivered through widely established top-down connections that inform low-level cortices about statistics represented at higher levels in the cortical hierarchy. While evidence supports that adaptation leads to priors reflecting the structure of natural images, it remains unclear if similar priors can be flexibly acquired when learning a specific task. To investigate this, we built a generative model of V1 that we optimized for performing a simple discrimination task and analyzed it along with large scale recordings from mice performing an analogous task. In line with recent successful approaches, we assumed that neuronal activity in V1 can be identified with latent posteriors in the generative model, providing an opportunity to investigate the contributions of task-related priors to neuronal responses. To obtain a normative test bed for this analysis, we extended the VAE formalism so that a task can be flexibly and data-efficiently acquired by reusing previously learned representations. Task-specific priors learned by this Task-Amortized VAE were used to investigate biases in mice and model when presenting stimuli that violated the trained task statistics. Mismatch between learned task statistics and incoming sensory evidence showed signatures of uncertainty in stimulus category in the posterior of TAVAE, reflecting properties of bimodal response profile in V1 recordings. The task-optimized generative model could account for various characteristics of V1 population activity, including within-day updates to the population responses. Our results confirm that flexible task-specific contextual priors can be learned on-demand by the visual system and can be deployed as early as the entry level of the visual cortex.

## 1 Introduction

Deep learning models, including discriminative and generative models, have been shown to successfully model neuronal responses in the visual system of the brain (Khaligh-Razavi & Kriegeskorte, 2014; Yamins & DiCarlo, 2016; Lotter et al., 2020; Csikor et al., 2025; Zhuang et al., 2021). These models were assessed through the efficiency of predicting neuronal responses to natural images or natural videos. However, visual cortical responses are not only determined by the stimulus itself but also by non-stimulus attributes, such as the the task the visual system is faced with (De Lange et al., 2018; Lange & Haefner, 2017; 2022). Notably, recent studies have demonstrated strong and systematic task-specific biases as early as the earliest stage of the visual cortex, the V1 (Corbo et al.,

---

*These authors contributed equally.

2022; 2025). Understanding these systematic biases requires that we understand the computational principles behind the changes occuring at the early stages of processing when learning a novel task.

A coherent normative framework for biological perception is probabilistic inference: learning is assumed to deliver a generative model of the environment, in which latent generative factors are inferred when a stimulus is observed (Yuille & Kersten, 2006; Fiser et al., 2010). In this context, neuronal activity in the visual cortex is interpreted as a representation of the posterior either by establishing a point estimate (Olshausen & Field, 1996; Schwartz & Simoncelli, 2001), approximating it through sampling (Lee & Mumford, 2003; Orbán et al., 2016), or through variational approximation (Ma et al., 2006), and biases can be formalized through learned priors. The entry stage to the visual cortex, the V1, learns elementary features of the environment (Hubel & Wiesel, 1959), suggesting a latent representation that has a close to linear relationship with the stimulus, i.e. suggesting a linear generative model underlying inferences (Olshausen & Field, 1996). The hierarchy of visual cortical regions permits a prior over the features represented in V1, as hierarchically higher layers that represent more complex statistics can establish contextual priors for lower layers through top-down connections (Lee & Mumford, 2003). Indeed, animal experiments established that higher level cortical activity delivers rich, structured influence on V1 (Lee & Nguyen, 2001; Chen et al., 2014; Kok et al., 2016; Ziemba et al., 2019) and these influences were shown to be aligned with the contextual priors acquired by a hierarchical generative model trained end-to-end on natural images (Csikor et al., 2025).

While these contextual priors reflect the regularities of the natural environment, these do not encompass more specific regularities that can arise when learning a task. In fact, it remains an open question if task-related contextual priors can be established in V1. In this paper we test the hypothesis that inference in a generative model that acquires task-specific priors underlies systematic biases in the V1 of task-trained mice.

To build a model of V1 that is primarily shaped by natural image statistics but is adapted to a task, we adopt a Variational Autoencoder approach (Kingma & Welling, 2013; Rezende et al., 2014) as these can perform flexible inference and thus provide an opportunity to investigate task-induced biases in the posterior. In this model, the latent variables of a Variational Autoencoder are identified with neurons of V1. Correspondence between the experiment and the model is established through matching the response properties of biological and model neurons. Latent activations are assumed to correspond to membrane potentials of neurons and firing rates are obtained by calculating the magnitude of responses (absolute value). We train the V1 model of VAE on natural images. To obtain representations matching that of V1, this VAE is constrained to have a linear generative model (Geadah et al., 2024), and use an extension that both produces more consistent performance in inference and better matching of contrast-dependent V1 responses (Catoni et al., 2024). Starting from the VAE model of V1, we extend the VAE formalism to adopt specific tasks, by adapting the variational posterior. In a standard approach, to obtain a new amortized variational posterior, the VAE needs to be retrained. This is neither efficient in terms of data requirements, nor biologically plausible, as it could steer away from a crucial representation obtained during the animal's development. Instead, we are seeking a way to build upon a learned representation when adapting to a new context. We propose a principled extension of the VAE formalism that is capable of flexibly learning contextual priors. The Task-Amortized VAE (TAVAE) reuses the likelihood of the task-general VAE and obtains task-specific posteriors without retraining the original amortized posterior. We train the TAVAE on a discrimination task matching a task mice were trained on while recording V1 activity (Corbo et al., 2022; 2025) to assess the predictions of TAVAE about response biases. Exploring the biases of a learned prior is made possible by the specific experimental design. Extensive training ensures that the animals learn the simple regularities of the task. In post-training test sessions, however, the learned stimulus distribution is systematically violated, exposing the effects of task-specific (contextual) prior. V1 neural activity is recorded through calcium imaging in 10 mice across 6 sessions, yielding recordings from 15,027 neurons while the animals perform the discrimination task.

In this paper, we first introduce the theoretical background for TAVAE. Second, we describe the animal experiment along with the specific generative model for V1. Third, we investigate the basic properties of a task-trained contextual prior and its consequences on population responses in V1. Fourth, we identify a signature of uncertainty about potential choices, multimodal responses emerging as a consequence of a mismatch between stimulus and the contextual prior, which we identify in population responses. Fifth, we investigate how updating the contextual prior reshapes response biases and show qualitative agreement with the transformation of population responses within an

experimental day when a new stimulus is introduced. Finally, we illustrate how the contextual prior and the stimulus likelihood can be inferred from V1 population responses and show that the inferred likelihood resembles the population activity recorded in animals with no exposure to the discrimination task.

## 2 THEORY

**Theory of Task-Amortized VAE.**  If a latent-variable generative model (e.g. a VAE) is trained on natural images, it can acquire a representation that reflect useful features of this environment. A VAE also learns a powerful tool for inference, the amortized variational posterior. When learning a task, the learned features can be useful for task execution even though the learned features do not exactly match the task variables. For instance, learning about natural images yields a Gabor filter-like representation, which can be easily queried about stimulus orientation. Here we develop a formalism, TAVAE, for task learning that capitalizes on the learned representation and also reuses the variational posterior. This model represents task-specific structure in a task-adapted prior. The formalism permits learning a prior through observing a limited number of observations from the task being performed. Here we apply the formalism to a simple setting where the V1-like representation is used to learn an orientation discrimination task.

Given a distribution of images, $p_0(\boldsymbol{x})$, we want to learn about this distribution by learning a latent representation $\boldsymbol{z}$:

$$p_0(\boldsymbol{x}) = \int p(\boldsymbol{x} \mid \boldsymbol{z})p_0(\boldsymbol{z}) \, d\boldsymbol{z}, \tag{1}$$

In general, inference about latent features, $p(\boldsymbol{z} \mid \boldsymbol{x})$, in such a model is intractable. Variational Autoencoders (VAEs) (Kingma & Welling, 2013; Rezende et al., 2014) have been proposed to provide an approximate posterior, $q(\boldsymbol{z} \mid \boldsymbol{x})$, relying on a variational approximation. For this, a lower bound to the log empirical distribution is optimized, called the evidence lower bound (ELBO). Through the optimization of the ELBO a pair of models is learned: the likelihood, $p(\boldsymbol{x} \mid \boldsymbol{z})$, termed the generative model, and the (amortized) variational posterior, $q(\boldsymbol{z} \mid \boldsymbol{x})$, termed the recognition model.

We seek to establish a method to perform inference in a computationally efficient way even if the data generating distribution changed from the natural distribution $p_0(\boldsymbol{x})$ to a task distribution $p_T(\boldsymbol{x})$. By default, adapting a VAE to new data requires retraining both the generative and recognition models from scratch by re-optimizing the ELBO. However, in many cases variation in the new dataset that occurs in the task is limited, thus the efficiency of optimizing the original set of parameters is also limited. Instead, we seek a computationally efficient way of adapting to different tasks by reusing the originally learned neural networks. We propose that the latent representation learned by the original VAE is retained as these learned latents can establish a useful feature space for the task. Consequently, we propose that the new generative model retains $p(\boldsymbol{x} \mid \boldsymbol{z})$ in the new context and the change in the stimulus distribution is solely induced by change in the latent prior $p_T(\boldsymbol{z})$:

$$p_T(\boldsymbol{x}) = \int p(\boldsymbol{x} \mid \boldsymbol{z})p_T(\boldsymbol{z}) \, d\boldsymbol{z}. \tag{2}$$

The task prior, $p_T(\boldsymbol{z})$, can also be thought of as the marginal of higher level latents in a hierarchical model, e.g. a mixture of options, o, and thus $p_T(\boldsymbol{z}) = \sum_{\text{o}} p_T(\boldsymbol{z} \mid \text{o}) \, p(\text{o})$. In this generative model, the new posterior is obtained by applying Bayes rule:

$$p_T(\boldsymbol{z} \mid \boldsymbol{x}) = \frac{p_0(\boldsymbol{z} \mid \boldsymbol{x})p_T(\boldsymbol{z})}{p_0(\boldsymbol{z})} \frac{1}{N(\boldsymbol{x})}, \tag{3}$$

where $N(\boldsymbol{x})$ is a normalization constant We can approximate the true posterior, $p_0(\boldsymbol{z} \mid \boldsymbol{x})$ in Eq. 3 with the variational posterior, $q(\boldsymbol{z} \mid \boldsymbol{x})$ coming from the trained VAE.

Eq. 3 highlights that once we know the prior related to the task, one can use the natural variational posterior to obtain the posterior under the prior associated to the task. To achieve this, we seek to maximize the log-likelihood under the latent prior, $p_T(\boldsymbol{z})$ using observations from the task $X_T$:

$$L = \sum_{\boldsymbol{x} \in X_T} \log(p_T(\boldsymbol{x})) = \sum_{\boldsymbol{x} \in X_T} \log \int d\boldsymbol{z} \, p(\boldsymbol{x} \mid \boldsymbol{z}) \, p_T(\boldsymbol{z}) \tag{4}$$

Details of the calculations can be found in Appx A.1.

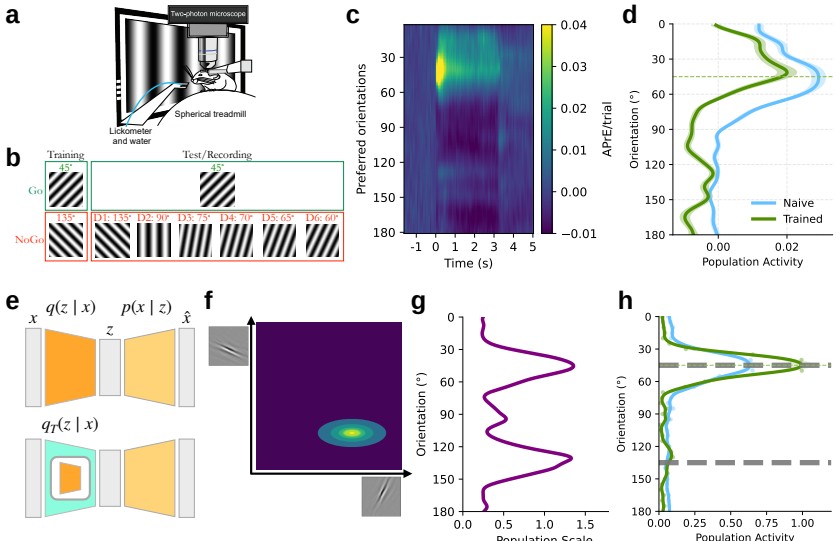

Figure 1: **a**, Experimental setup. **b**, Go-NoGo visual discrimination task stimuli with training and testing schedule. **c**, Population response map of the recorded V1 population on D1 for the Go stimulus. **d**, Population response profile averaged over time for trained and untrained (naive) mice. **e**, Cartoon of VAE (top) and TAVAE (bottom) models. The TAVAE generative model component of VAE (*pale orange*) is reused in TAVAE whereas the recognition model of VAE (*dark orange*) is transformed through reweighing the variational posterior with the task-prior (Eq. 3). **f**, Illustration of the posterior of two latent dimensions; inset: receptive fields of latents. **g**, Task prior of TAVAE for the discrimination task. Scale of the Laplace prior is shown. **h**, TAVAE responses to the Go stimulus with task (green) and natural priors (blue).

**VAE model of V1.** To be able to test the predictions of task-related contextual priors in biological recordings, we first built a VAE end-to-end trained on natural images that reflected basic properties of V1. In our study we take a normative approach by seeking correspondence between a generative model of a task-adapted model V1 and neural recordings from V1. This way, optimization is based jointly on natural image statistics and task structure but not on neural recordings.

Latent variables, $z$, of the generative model were identified with neurons of V1. This is motivated by the similarity of learned representations to the sensitivities of neurons in V1 (Geadah et al., 2024; Catoni et al., 2024; Olshausen & Field, 1996). Before specifying the model, we elaborate on the correspondence between the VAE and neural activity. Upon presenting an image, the VAE returns a variational posterior. Many early accounts identified neuronal responses with the MAP estimate (Olshausen & Field, 1996; Schwartz & Simoncelli, 2001), which was justified by the fact that neural recordings report response means. Alternatively, the responses of neurons can be identified with a sampling approximation of the posterior, which assumes that temporal variations in responses correspond to sample variance from the posterior (Hoyer & Hyvärinen, 2002; Orbán et al., 2016; Echeveste et al., 2020; Festa et al., 2021; Shrinivasan et al., 2023). We do not commit to a particular representation at this point, but distinguishing the two can be the subject of later investigations.

We trained a VAE on natural image patches. These patches were 40-pixel whitened crops from the van Hateren database (Van Hateren & van der Schaaf, 1998). The VAE had four characteristic features. 1, Following an earlier VAE study (Geadah et al., 2024), the generative model was linear, analogous to independent component accounts of V1 organization (Olshausen & Field, 1996); 2, The latent space was large dimensional (1799 dimensions) mimicking the complete / overcomplete structure of V1 and ensuring that the linear generative model can accommodate the information in the observations; 3, The prior over latents was Laplace, motivated by efficient coding considerations, and also aligned with earlier accounts (Olshausen & Field, 1996). This condition is necessary to match the latents of the VAE with V1 neurons as this ensures learning Gabor-like receptive fields, a key property of V1 neurons (Csikor et al., 2022); 4, A scaling latent was introduced, mimicking the Gaussian Scale Mixture (GSM) family of models of natural images (Wainwright & Simoncelli,

1999). This choice is motivated by the joint statistics of natural images, the superior performance of GSM's to predict V1 responses, and an earlier study which showed that this extended version of VAE ensures more reliable inference, especially in lower contrast regimes (Catoni et al., 2024) (in Appx. B.9 we compare the main results with a model based on a VAE without the scaling factor). In summary, the generative model of this baseline model was $p_\varphi(\boldsymbol{x} \mid \boldsymbol{z}, s) = \mathcal{N}(\boldsymbol{x}; \exp(s) \cdot \boldsymbol{A}\boldsymbol{z}, \sigma^2\boldsymbol{I})$, $p_0(\boldsymbol{z}) = \text{Laplace}(\boldsymbol{z}; 0, 1)$, $p(s) = \mathcal{N}(s; 0, 1)$, where $\exp(s)$ ensures that the scaling is positive. Based on this generative model, we optimize the negative ELBO:

$$\mathcal{L}(\mathbf{x}, \varphi, \theta, \psi) = -\mathbb{E}_{q_\theta(\boldsymbol{z}|\boldsymbol{x})q_\psi(s|\boldsymbol{x})}\left[\log p_\varphi(\boldsymbol{x}|\boldsymbol{z}, s)\right] + D_{KL}(q_\theta(\boldsymbol{z}|\boldsymbol{x})||p(\boldsymbol{z})) + D_{KL}(q_\psi(s|\boldsymbol{x})||p(s)),$$
(5)

such that the variational posteriors $q_\theta(\boldsymbol{z} \mid \boldsymbol{x})$ and $q_\psi(s \mid \boldsymbol{x})$ were modeled as Laplace and Normal distributions, respectively (we will omit the subscript from now on). The latent variables of this VAE can take positive and negative values too. Similar to earlier accounts, $\boldsymbol{z}$ is interpreted as membrane potential responses of biological neurons (Orbán et al., 2016; Echeveste et al., 2020). To obtain firing rate-like responses, we use the response magnitude (absolute value) of latent activations when comparing to biological data, similar to Csikor et al. (2025); Geadah et al. (2024). In contrast with applying a ReLU for positive outputs, this avoids clipping the representational space.

**Orientation discrimination task for mice and machine.** We characterized the biases emerging after adaptation to a task in mice and in TAVAE model by studying an orientation discrimination task. For this we used identical task stimuli in experiment and model.

Mice performed a visual Go-NoGo task in which animals were required to lick for a $45°$ or withhold from licking for a $135°$ moving grating stimulus (Fig. 1a) (Corbo et al., 2025). Animals were first trained on the orientation discrimination task until reaching proficient performance, after which they underwent a six-day testing period accompanied by neural population recordings. During the testing period the Go stimulus remained identical to the training phase but the NoGo stimulus systematically violated the trained task statistics. Specifically, the NoGo stimulus was progressively shifted toward the Go orientation, reducing the angular difference to $15°$ by Day 6 (Fig. 1b).

We used a pair of oriented grating stimuli to train the prior of a TAVAE (in the baseline experiment the pair of orientations were $45°$ and $135°$). In order to emulate moving gratings, each oriented grating was presented at 50 different phases. Response of model neurons was established as the average of the absolute value of the posterior means.

We trained task-specific priors on this stimulus set. The task prior, $p_T(\boldsymbol{z})$, corresponds to a mixture of the the two stimulus categories and we seek to learn the marginal of this distribution (Eq. 2). Acquiring the task was modeled by learning this marginal prior, $p_T(\boldsymbol{z})$, according to Eq. 12 (Fig. 1e).

To learn the approximate posterior for the (discrimination) task, we applied the TAVAE formalism. We approximated the prior with a diagonal covariance matrix. Since the variational posterior (and also the prior) is factorized $q(\boldsymbol{z}, s \mid \boldsymbol{x}) = q(s \mid \boldsymbol{x}) \cdot q(\boldsymbol{z} \mid \boldsymbol{x})$ over s and $\boldsymbol{z}$, Eq. 3 translates to:

$$q_T(\boldsymbol{z}|\boldsymbol{x}) \propto \frac{q(\boldsymbol{z}|\boldsymbol{x}) \, p_T(\boldsymbol{z})}{p_0(\boldsymbol{z})},$$
(6)

where $p_T(\boldsymbol{z})$ was the prior over latents defined by the task. We search for an appropriate task prior as a zero mean Laplace distribution. The choice of constraining the mean to zero is motivated by a symmetry of $\boldsymbol{z}$ activations around zero across all the phases of the grating stimulus:

$$p_T(\boldsymbol{z}) = \text{Laplace}(\boldsymbol{z}; \boldsymbol{0}, \underline{\sigma}_T) = \prod_{i=1}^{N} \frac{1}{2\sigma_{T,i}} \exp\left(-\frac{|z_i|}{\sigma_{T,i}}\right).$$
(7)

Under these settings, training the task prior amounted to learning the variances, $\underline{\sigma}_T$. We show in Appx A.2 that the minimization of the general task loss Eq. 4 simplifies to solving the self-consistency equation (see B.6 as well):

$$\sigma_{T,i} = \frac{1}{n} \sum_{\boldsymbol{x} \in X_T} \mathbb{E}_{q_{T,\sigma}(z_i|\boldsymbol{x})}\left[|z_i|\right].$$
(8)

In summary, the model of task-adapted V1 is trained jointly on natural images and the task stimuli but it was not directly optimized to fit neuronal responses.

**Characterization of responses in mice and model.** We analyzed calcium imaging recordings from excitatory neurons in layer 2/3 of mouse V1 during the test phase of an orientation discrimination task. Neural activity was summarized using a population response profile, in which neurons are ordered by their preferred orientation (measured in a separate tuning block) and responses are averaged across time and neurons. Population profiles were pooled across animals to reduce recording noise and are visualized with error bars denoting 95% confidence intervals of the mean, as specified in the figure captions (see Appendix B.3 for details).

To obtain an analogous measure in the model, we identified each latent variable $z$ with a putative V1 neuron and estimated its orientation tuning curve using responses to grating stimuli. The majority of latent components were orientation selective. Latent variables were ordered by their preferred orientation and population response profiles were computed using $5°$ orientation bins. Variability across model neurons is shown as 95% confidence intervals estimated by bootstrap resampling. Full methodological details are provided in Appendix B.4.

To quantify the predictive power of the model, we calculated the Pearson correlation coefficient ($r$) between the experimental and model population response profiles in Table 1 (aggregated in $5°$ bins, as in Fig. 3). To specifically assess the effect of the task, we calculated the 'contextual modulation' of the response profile, the difference in response profiles of task-engaged and naive animals, and analogously, as the difference between response profiles of TAVAE and VAE (no prior change). We also performed neural predicitivty and Centered Kernel Alignment analysis for across-image alignment of model and experimental neural responses (Appx. B.10).

## 3 RESULTS

**Effect of contextual prior for task-congruent stimuli.** First, we tested how task-specific training influenced model inference and compared the effects of contextual priors in task-trained versus naive animals. We first focused on the first recording session (D1), in which the test stimuli matched those used during training. To align the model with the experimental paradigm, we synthesized $45°$ and $135°$ gratings; we varied phases (Appx. B.1) to emulate motion, and fit the task prior using Eq.8. The optimization converged after five iterations (Appx. Fig.7). Using the acquired prior, we computed tuning curves and subsequently population response profiles. The tuning curves showed systematic modulation: neurons whose preferred orientations flanked the orientations on which the prior was trained were suppressed (Appx. Fig. 6). Population response profiles for both trained stimulus orientations showed clear peaks (Fig. 2a,b). We used a reduced contrast stimulus for testing, in line with the experiment data (Corbo et al., 2025) (in Appx. B.9, we show the contrast dependence of some of our results).

To understand the effect of a contextual prior, we contrasted the task-optimized TAVAE response profiles with the task-general, standard VAE profiles (Fig. 2a,b). Similarly, we compared experimental response profiles recorded in sessions when the Go and NoGo stimuli were identical with training stimuli ($45°$ and $135°$, D1) with response profiles recorded in naive animals. The task-optimized contextual prior resulted in sharpening of the response profile (Fig. 2c, t-test, n=10 grating phases). Similar sharpening was evident in mice when analyzing population activity (Fig. 2g, t-test p<0.01, n=60, means from 6 days, two stimuli and autocorrelation-correcting to 5 independent timeframes). Sharpening in TAVAE can be understood as suppressing the responses of neurons with preferred orientations not matching the task orientation (Appx. Fig. 6).

Another signature of a task-optimized generative model is a significant reduction in baseline activity, which we measured by calculating the average activity in the $(0, 30)$ and $(150, 180)$ orientation window. Upon learning the task, the selective suppression of neurons that have preferred stimulus orientation not matching the stimulus (Appx. Fig. 6) results in a suppressed baseline (Fig. 2d, t-test n=12, means from 6 days, two stimuli). This suppression is consistent across stimulus conditions in the experiment (Fig. 2e,f,h) as well.

In the task-optimized TAVAE, suppression of baseline activity is not homogeneous: when presenting a task stimulus, suppression does not affect neurons tuned to training stimulus orientations (Fig. 2a). This can be explained by the heterogeneity of the scale of the task prior, $\underline{\sigma}_T$ (see Fig 2g, Fig 7): along directions where the prior is wider (the trained orientations), the posterior will be less affected, resulting in baseline activity similar to the task-general VAE. When considering experimental recordings, the magnitude of the heterogeneity in baseline suppression is close to the level of fluctuations.

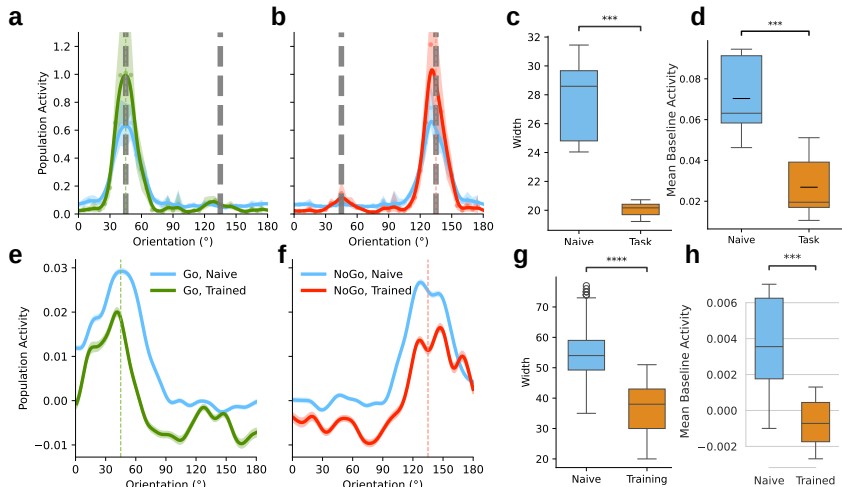

Figure 2: **Effect of learning the discrimination task in the model and in the experiment. a, b**, Response profiles to task stimuli (45° and 135° gratings, red and green dashed lines, respectively) using natural prior (blue) and the task prior (solid green and red for Go and NoGo respectively) in the model, smoothing 3°. Dashed grey lines correspond to the learned contextual prior. Shading: bootstrap estimate of 95% confidence interval of the mean across model neurons. **c**, Width of the 45° peak using the natural prior (blue) and task prior (orange; statistics are computed from different grating phases). **d**, Baseline reduction in TAVAE. Baseline activity was computed for each stimulus configuration from Fig. 2a,b and Fig. 3b. Box plot and significance markers are based on statistics computed across these stimuli. **e, f**, D1 Go and NoGo responses in naive (blue) and trained (green, red) mice, smoothing 6°, shades 2 sem. **g**, Width of the Go response over days and timeframes for naive (blue) vs. trained animals (orange, mean over neurons, trials). **h**, Baseline reduction following training in the experiment. Baseline activity was computed for each stimulus configuration from Fig. 2e,f and Fig. 3a. Boxplots: 25-75%, whiskers 1.5·IQR. Asterisks indicate statistical significance.

Although the predicted inhomogeneity in baseline suppression can occasionally be experimentally observed (see the bump around 135° in Fig. 2e), noise in recordings prevents a general conclusion.

**Systematic biases of the contextual priors for OOD stimulus.** Across five experimental sessions, animals were exposed to stimuli that systematically deviated from the trained stimulus set. More precisely, the NoGo stimulus gradually converged to the Go stimulus (Fig. 1b). Population response profiles from these sessions show drastic deviations from the response predicted merely by the stimulus orientation (Fig. 3a). In particular, the peak of the population response profile is distinct from the actual stimuli in multiple sessions across D3-D6, as measured by the mean absolute difference between peak estimate (200 × bootstrap 5 random cells per orientation, is higher for trained animals compared to naive, $p < 10^{-6}$, one-tailed t-test, see also Fig. 3a). Furthermore, multimodal responses are characteristic in multiple conditions (especially in D4 and D5). Counterintuitively, multimodal responses display troughs at the actual stimulus orientation, with peaks flanking the stimulus orientation from both sides (higher response at the bimodal peaks compared to the troughs in D4-D5: flanking orientation peaks to stimulus trough, $p < 10^{-6}$, one-tailed t-test; n ≈ 2000-20000, for the 5-20 cells in the respective 1° orientation bins, Fig. 3a).

We assumed that these biased responses are a consequence of the uncertainty about the source of the stimulus when the contextual prior does not match the observation. Indeed, when simulating experimental population responses across the five conditions (corresponding to experimental days) such that the contextual prior was matched to the stimuli (corresponding to the mice learning a new prior every day), no distortions were observed (Appx. Fig. 8a). Also, this 'eager adapter' version of TAVAE predicted experimental population response profiles poorly (see Table 1 for details).

To investigate alternative hypotheses, we assessed TAVAE responses under two additional assumptions. First, we assumed that the contextual prior acquired during the extensive training is retained, i.e. we assumed a prior at 45° Go and 135° Nogo stimuli (Appx. Fig. 8b). The population response profile showed very weak bimodality and, accordingly, showed relatively weak correlation

|  | TAVAE (45, 90) | VAE | TAVAE (45, 135) | TAVAE (EAGER) |
|---|---|---|---|---|
| $r$(TASK) | **0.78 ± 0.02** | 0.53 ± 0.12 | 0.54 ± 0.10 | 0.53 ± 0.11 |
| $r$(TASK–NAIVE) | **0.58 ± 0.09** | – | 0.32 ± 0.17 | −0.10 ± 0.23 |

Table 1: Correlation between model and experimental population activity across orientation bins for different TAVAE configurations and the naive VAE (averaged over stimuli). The standard error of the mean is also shown. Top row: correlation between task-engaged experimental data and model predictions. Bottom row: correlation of task-modulation of population response profile (difference between task-engaged and naive conditions) between model and experiment. The analysis was performed on orientation bins within the (30°, 105°) window, thereby excluding orientations far from the stimuli presented on those days where measurement noise is more dominant. For orientations distant from the stimulus, the model predicts primarily the baseline reduction shown in Fig. 2d,h.

between TAVAE and experimental results (Table 1). Second, we reasoned that the abrupt change in the NoGo stimulus from 135° to 90° on D2 likely induced an update in the prior. This interpretation is supported by the observation that the false alarm rate of the animal on Day 2 is comparable to that on Day 1 (Appx. Fig. 9a), indicating that the animal is adapting to the new NoGo stimulus. Notably, the false alarm rate further decreases across behavioral quintiles within Day 2 (Appx. Fig. 9b). In contrast, from Day 3 onward, the false alarm rate steadily increases (Appx. Fig. 9a) (indicating that the animal is not adapting further). Confirming this assumption, a prior reflecting 45° Go and 90° NoGo stimuli reproduces the systematic biases observed in the experiments (Fig. 3b): the population response peak shifts away from the actual NoGo stimulus, as observed experimentally during D3–D6, and bimodality is present, with a local minimum near the stimulus orientation, for the NoGo stimulus under the conditions used during D4. Indeed, for the shifted peak of the response profile we found that the absolute difference between the peak position and the stimulus orientation was significantly larger in the TAVAE than in the naive VAE for all stimuli except 90°, where no prior–stimulus mismatch was present (Appx. Fig. 10 and see Appx. B.8). To test whether the population response profile displayed bimodality with a dip at the orientation corresponding to the NoGo stimulus, we identified the positions of maximal population activity and compared to the activity at the stimulus orientation. For the 70° stimulus (corresponding to D4), the flank maxima exceeded the activity at the stimulus position on both sides (one-tailed t-test, left: $p = 0.001$; right: $p = 0.028$; $n_{stimulus} = 48$, $n_{left} = 32$, $n_{right} = 9$), indicating the presence of a local minimum between the two flanking orientations. Samples were drawn from individual model neuron responses within the orientation bins used for the comparison. Beyond the presence of signatures of systematic biases, we also assessed the performance of TAVAE in predicting the experimental population response profile. When assuming a 45° Go and 90° NoGo prior, the correlation between TAVAE prediction and experiments exceeded alternatives (Table 1), including the task-general VAE. We further characterized the alignment of model predictions with experimental recordings through Neural Predictability (Nayebi et al., 2023) and Centered Kernel Alignment (Kornblith et al., 2019) analyses (Appx. B.10). These analyses confirmed that TAVAE across-stimulus predictive power exceeds that of the task-general VAE (Tables 4, 3).

To explore the contribution of model choices in TAVAE, we performed experiments with altered contrast levels (Appx. Fig. 11) and ablating the scaling variable from the baseline model (Appx. Fig. 12, see also Appx. B.9). Analyses showed that the decreased contrast contributed to better model fits (Table 2). Intuitively, this lowered contrast increases uncertainty, which has a defining role in rendering alternative options similarly viable. Note, that lowered contrast was inspired by matching the experimental settings (Corbo et al., 2025). Our analyses also highlighted that the GSM-style scaling was contributing to slightly better model fits (Table 2) but integrating TAVAE with a VAE without scaling also delivered significant correlations with the experimental data.

**Signature of updating the contextual prior.** Our results indicate that D1 responses are consistent with a contextual prior peaking at 45° and 135°. Further analysis of D2 through D6 sessions indicated a different contextual prior peaking at 45 and 90°. Taking these findings together, animals seem to shift their priors from the first session (D1) when stimuli are identical with training stimuli to the second session when the NoGo stimulus is radically updated. Behavioral results indicate an adaptation during D2 (see the gradual reduction in the false alarm rate as the day progresses in Appx. Fig. 9b), motivating an analysis of population activity throughout the experimental session of D2.

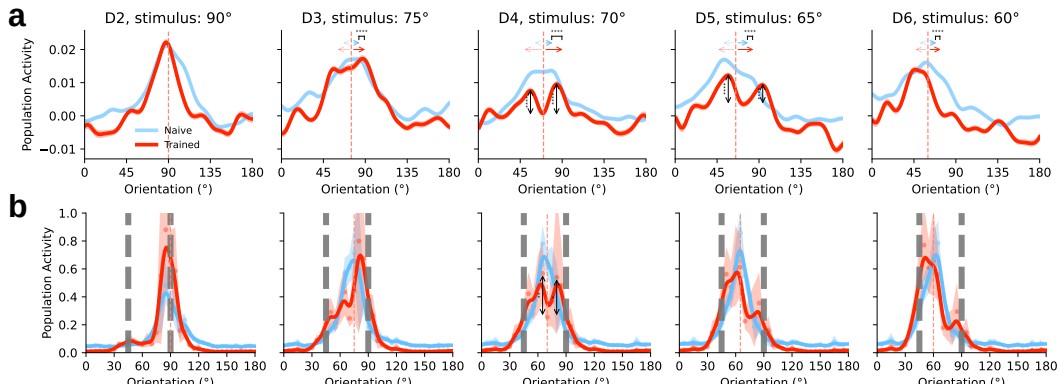

Figure 3: **Systematic biases for mismatched training and test in experiment and model**. **a**, V1 responses over five experimental sessions during which the NoGo stimulus (red) is deviating from the training NoGo stimulus at 135°, confidence shades 2 sem. Responses from naive animals are shown for reference (blue). Bimodal peaks-to-middle-trough activity differences (vertical black arrows). Mean absolute difference between stimulus and peak orientation (horizontal red and blue mirrored half arrows). **b**, Model responses on the same stimulus orientations as in the experiment across the five sessions. A prior learned for 45° and 90° gratings is assumed (grey dashed lines), bimodal peaks-to-middle-trough without smoothing (vertical black arrows). All panels: data points, smoothing and confidence intervals as in Fig. 2, asterisks indicate statistical significance (see text).

We model the shifting prior by fitting the task prior to a dataset with gratings characterized by unbalanced orientation classes. Namely, we investigated a dataset that contains 45°, 90°, and 135° gratings with the ratio $1 : \gamma : 1-\gamma$. We obtained contextual priors for $\gamma \in \{0.1, 0.25, 0.5, 0.75, 0.9\}$ and calculated population response profiles to the D2 NoGo stimulus, oriented at 90°.

Population response profile of high-weight 135° priors (i.e. $\gamma < 0.5$) reveals two additional modes flanking the central stimulus-aligned mode (Fig. 4a). These correspond to (shifted) modes contributed by the alternative hypotheses, i.e. that the stimulus is coming from the 45° and 135° components of the prior. A low weight of the 135° stimulus (high $\gamma$) in the contextual prior results in asymmetric flanking modes whereby the mode corresponding to the 45° component is greater than the 135° (Fig. 4b). For the example pair of $\gamma$s (0.25 and 0.75, the asymmetry is found to be greater at $\gamma = 0.75$ than for $\gamma = 0.25$ (t-test, $p < 10^{-6}$, $n = 806$; sample pairs were constructed by sample individual model neurons from the two bins). In fact, if the prior is updated from D1 to D2, we expect a gradual decline in the flanking peak closer to the NoGo stimulus.

To test this prediction in the data, we stratified D2 trials into five quintiles and analyzed population response profiles separately. Flanking modes were consistently present both early and late in the session (Fig. 4c). Further confirming our hypothesis, the late trials display asymmetric flanking modes. Namely, as the session progresses from early trials (quintile 1, Q1) to late trails (Q5), the difference between Go and NoGo flanking peaks widen as shown in Fig.4c,d (t-test, $p < 10^{-6}$ with n=1685 equal random sample at the 45° and 135° orientation bins, with number of trials 10-30). This shift is also reflected behaviorally by a gradual reduction in false alarm rates across quintiles (Appx. Fig. 9b).

Importantly, as expected from theory, the flanking modes are 'attracted' towards the orientation of the NoGo stimulus, i.e. to 90°, in line with the formation of an orientation posterior by combining a contextual prior over orientations with a stimulus-driven likelihood. We argue that shifts of 45° and 135° modes towards the NoGo stimulus on D2 is a signature of probabilistic inference under uncertainty: a wide likelihood combined with a finite-width prior results in a shifted posterior. We then use this insight to estimate the likelihood and the orientation prior from the position of the modes from the population response profiles (see Appx. B.11 for derivation). The inferred likelihood is substantially wider than the orientation prior (Fig. 4e,f, F-test p<$10^{-6}$), which is consistent with the relatively small shift of the modes away from the originally trained stimulus orientations. This wide likelihood resembles the population response profile observed in untrained animals (F-test p>0.13). In untrained animals, where the orientation prior is effectively flat and cannot sharpen the posterior, the population response profile provides a close approximation of the likelihood itself.

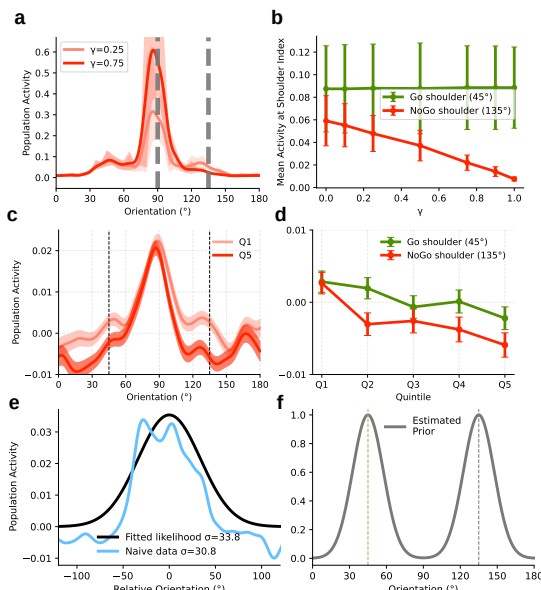

Figure 4: **Signatures of updating a contextual prior**. **a**, TAVAE population response profiles under a mixture contextual prior, where the weight ($\gamma$) of the 90° components is increased from 0.25 to 0.75. **b**, Progression of the heights of flanking modes with changing mixture components simulating the progression of the trials across the 90° stimulus session for TAVAE. Error bars: 95% CI for the mean estimate **c**, Population response profile on D2 when 90° stimulus is introduced, for early (Quintile 1) and late trials (Quintile 5). **d**, as *b* but for experiment on D2. *a-d*: smoothing, confidence intervals as in Fig. 2. **e**, Orientation likelihood inferred from D2 Q1 population response profile (black, centered on 0°) and population response profile of naive animals (blue, centered on 0°). **f**, D2 prior inferred from population response profile.

## 4    DISCUSSION

Here we investigated the biases that emerge from task learning in a generative model and compared them to the biases observed in animals extensively trained on the same task. For this, we developed a variant of VAEs, the TAVAE in which the prior can be flexibly adapted in a computationally efficient way. This permitted the reuse of the representation learned for natural images. TAVAE could account for a range of phenomena, including the sharpening of population responses with learning (without changing the receptive fields of neurons) and changes in baseline. Crucially, the model revealed bimodal responses, a hallmark of probabilistic inference under uncertainty, and this bimodality tightly aligned with that observed in V1 when trained stimuli were not matched with actual observation. TAVAE could also capture the updating of the prior when the animal was faced with updated task contingencies.

TAVAE permits the explicit reuse of the recognition model when acquiring a new task. Although we do not argue that this algorithmic solution is the one used by the neural circuitry, it provides insights into how the representation is reshaped through learning. The systematic distortions of neuronal responses were reported from layer 2/3 neurons, where bottom-up and top-down signals are integrated. The laminar organization of the cortex might permit more complex computations, for instance layer 4 neurons might actually preserve the original recognition model to be modulated by top-down influences when the feed-forward input reaches layer 2/3. It is a delicate question how a task-related contextual prior is integrated with the contextual priors reflecting the regularities of natural images (Csikor et al., 2025). The representational geometry of population responses might give a clue about this (Lazar et al., 2024) but this remains an open question.

We chose the task prior as a simple modification of the natural prior, namely by retaining independence among latents and tuning only their variances. Interestingly, although the posterior in the high-dimensional latent space was unimodal, this still translated into a multimodal population response profile in the one-dimensional orientation space within a certain parameter range. The TAVAE formalism is capable of accommodating more complex priors with more flexible form. In particular, different modes in the latent space might correspond to different options, yielding multimodal posteriors. Such priors can be learned through the posteriors associated with different options. In principle, the framework can also be applied to model contextual modulations in higher cortical areas. More broadly, deep generative models may become an important tool for modeling cortical activity under realistic conditions.

**Reproducibility statement.** The mathematical derivation needed for the results can be found in the main text 2, and in the appendix A.1, A.2, B.5. The code used for modeling can be found in the Supplementary material. In the code, we provided a script that downloads the model weights of the VAE used in the paper. Mouse V1 activity data can be downloaded from https://zenodo.org/records/8109858, and used with the data analysis scripts: https://github.com/CSNLWigner/mouse-V1-task-priors.

**Acknowledgments.** This work was supported by the European Union project RRF-2.3.1-21-2022-00004 within the framework of the Artificial Intelligence National Laboratory in Hungary (GO), the National Research, Development and Innovation Office under grant agreement AD-VANCED 150361 (GO), and the Fulbright U.S. Student Program (KM). The authors thank HUN-REN Cloud for providing compute support for the study (see Héder et al. (2022); https://science-cloud.hu/). We also thank anonymous ICLR reviewers for their constructive feedback and helpful suggestions.

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

## A APPENDIX

### A.1 LEARNING THE TAVAE PRIOR

We derive an optimization objective to determine the task prior. We would like to maximize the log-likelihood under the latent prior, $p_T(\boldsymbol{z})$ using observations from the task $X_T$:

$$L = \sum_{\boldsymbol{x} \in X_T} \log(p_T(\boldsymbol{x})) = \sum_{\boldsymbol{x} \in X_T} \log \int d\boldsymbol{z}\, p(\boldsymbol{x} \mid \boldsymbol{z})\, p_T(\boldsymbol{z}) \tag{9}$$

If we assume that the variational posterior of the natural images is a good approximation of the true posterior

$$p(\boldsymbol{x} \mid \boldsymbol{z}) \approx \frac{q(\boldsymbol{z} \mid \boldsymbol{x})\, p_0(\boldsymbol{x})}{p_0(\boldsymbol{z})}, \tag{10}$$

then the log-likelihood, with $p_0(\boldsymbol{x})$ moved outside of the $d\boldsymbol{z}$ integral, can be written as:

$$L = \sum_{\boldsymbol{x} \in X_T} \left[ \log p_0(\boldsymbol{x}) + \log \int d\boldsymbol{z}\, \frac{q(\boldsymbol{z} \mid \boldsymbol{x})\, p_T(\boldsymbol{z})}{p_0(\boldsymbol{z})} \right] \tag{11}$$

As the first term does not depend on the task prior, the functional to maximize is:

$$L' = L - L_0 = \sum_{\boldsymbol{x} \in X_T} \log \frac{p_T(\boldsymbol{x})}{p_0(\boldsymbol{x})} = \sum_{\boldsymbol{x} \in X_T} \log \int d\boldsymbol{z}\, \frac{q(\boldsymbol{z}|\boldsymbol{x})p_T(\boldsymbol{z})}{p_0(\boldsymbol{z})}, \tag{12}$$

where $L_0$ is the log-likelihood of the task data under the original model.

### A.2 LEARNING THE PRIOR FOR THE DISCRIMINATION TASK

As explained in the main text, we seek the task prior in the form of a zero mean Laplace distribution:

$$p_T(\boldsymbol{z}) = \text{Laplace}(\boldsymbol{z}; 0, \underline{\sigma}_T) = \prod_{i=1}^{N} \frac{1}{2\sigma_{T,i}} \exp\left( -\frac{|z_i|}{\sigma_{T,i}} \right). \tag{13}$$

The scales can be determined in a principled way by taking the derivative of the log-likelihood with respect to the scales.

$$0 = \frac{\partial L'}{\partial \sigma_{T,i}} = -\frac{n}{\sigma_{T,i}} + \sum_{\boldsymbol{x} \in X_T} \frac{1}{\sigma_{T,i}^2} \frac{\int dz_i\, \frac{q(z_i \mid \boldsymbol{x})}{p_0(z_i)} |z_i| \exp\left( -\frac{|z_i|}{\sigma_{T,i}} \right)}{\int dz_i\, \frac{q(z_i \mid \boldsymbol{x})}{p_0(z_i)} \exp\left( -\frac{|z_i|}{\sigma_{T,i}} \right)}, \tag{14}$$

where $n$ is the number of images in the task dataset. This can be rearranged to an intuitive form:

$$\sigma_{T,i} = \frac{1}{n} \sum_{\boldsymbol{x} \in X_T} \mathbb{E}_{q_{T,\sigma}(z_i|\boldsymbol{x})} \left[ |z_i| \right] \tag{15}$$

This is a self-consistency equation, since, of course, the right-hand side also depends on $\sigma_i$. We solve this iteratively. We start with the original prior $p_T^{(0)}(\boldsymbol{z}) = p_0(\boldsymbol{z})$. Then after the first iteration, the scales for each dimension will correspond to the scale from the variational posterior, which will be refined in the subsequent iterations. The resulting approximate posterior (Eq. 6) is no longer a Laplace distribution but a piecewise exponential function. Because of this, the necessary integrals can be performed analytically to calculate the various moments, as we demonstrate in Section B.5.

# B EXPERIMENTAL ORIENTATION DISCRIMINATION TASK AND ITS VAE IMPLEMENTATION

## B.1 GRATINGS DATASET

We generated a synthetic dataset consisting of $40 \times 40$ grayscale images of sinusoidal gratings. Each image is defined by the function

$$g(X,Y) = C \cdot \sin\left(2\pi f \cdot \left(X \cos(\theta) + Y \sin(\theta)\right) + \phi\right),$$

where $f$ represents the spatial frequency (fixed at 3), $\theta$ is the orientation angle (measured in radians), $\phi$ stands for the phase, and $C$ serves as the contrast scaling factor.

The coordinates $(X,Y)$ correspond to pixel locations on a uniform $40 \times 40$ grid spanning the interval $[-1,1]$ in both dimensions. For fitting the scales of the task prior, we selected a contrast factor of 1, while for inputting the test stimuli into the model, we opted for a contrast factor of 0.3 to simulate the reduced contrast used during the test phase of the experiment. The dataset comprised 36 angles spanning from 0° to 180°, along with 50 distinct phase values.

## B.2 RECEPTIVE FIELDS AND TUNING CURVES OF EAVAE AND TAVAE

The latent variables $z$ of our baseline VAE model EAVAE have a linear receptive field (determined by the image produced when just one $z$ value is assigned a 1 and all others are set to zero) consist of localized oriented filters (see Fig. 5 a).

Thus, it is reasonable to infer that (some) of these model neurons demonstrate orientation selectivity when gratings at different angles pass through the network, as illustrated by the examples in Fig. 5b.

Priors affect the tuning curves. In Fig. 6, we show that neurons whose preferred orientations flank the prior orientation are suppressed, as indicated by a decrease in their tuning curve amplitude.

## B.3 EXPERIMENTAL DATA

Calcium imaging recordings were obtained from excitatory neurons in layer 2/3 of mouse V1 during the test phase of the experiment. Recordings were performed in 10 mice expressing GCaMP6f or GCaMP6s, across six experimental sessions corresponding to six stimulus conditions. Each session contained between 2007 and 2675 neurons and 50–150 trials.

Calcium signal preprocessing followed the procedure described in Corbo et al. (2025). Fractional fluorescence ($\Delta F/F$) signals were baseline-corrected using the median fluorescence during the preceding intertrial interval and deconvolved to infer action potential–related events (APrEs), which served as a proxy for spiking activity.

Neural responses were computed by averaging activity over 40 frames corresponding to the central 1–1.5 s of stimulus presentation. Temporal autocorrelation in calcium signal time-courses was accounted for by correcting the effective sample size to $n = 7$ when computing standard errors of the mean (s.e.m.) and performing statistical tests.

Population response profiles were constructed by arranging neurons according to their preferred orientation, estimated from a separate tuning block and averaging responses across neurons and

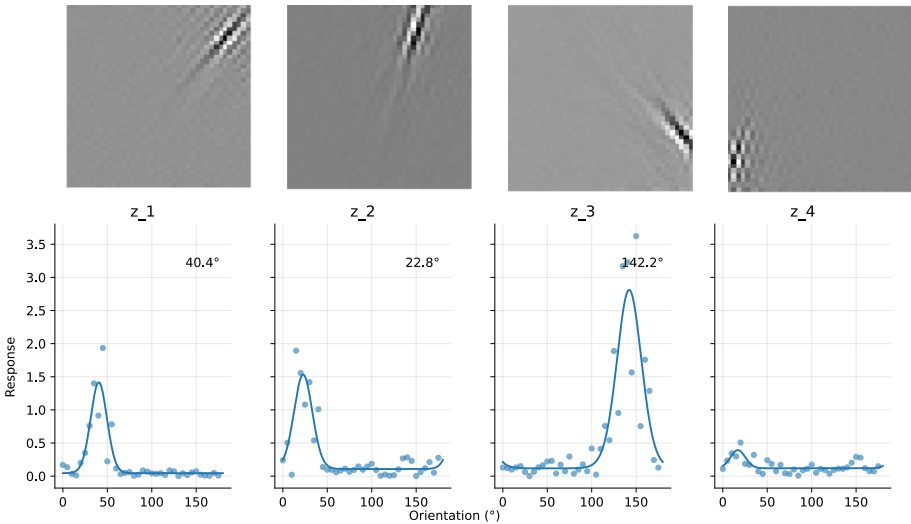

Figure 5: **a**, Receptive fields for example model neurons in EAVAE base model. **b** Tuning curves and von Mises fits for these neurons. Note that the last neuron is classified as not orientation selective.

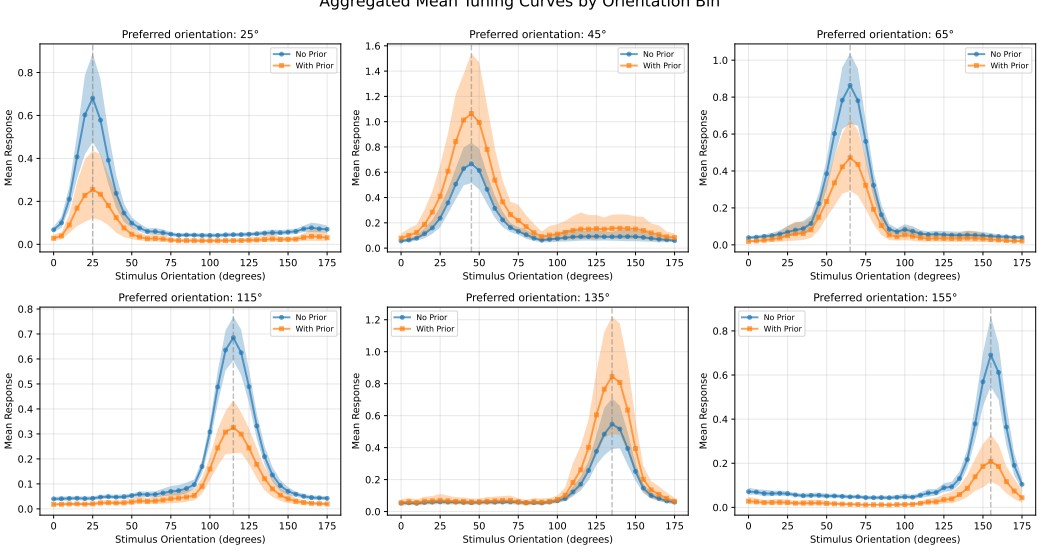

Figure 6: Tuning curve modulation of individual model neurons (aggregated by preferred orientation) for the naive model (blue) and the model trained with priors on 45° and 135° grating stimuli (orange). Neurons whose preferred orientations flank the prior stimulus directions are suppressed. Shading: 95% CI of the mean.

time. Profiles were pooled across animals to reduce recording noise, resulting in approximately 2–20 neurons per 1° orientation bin. Error bars and shaded regions denote either $\pm 2$ s.e.m. or $95\%$ confidence intervals of the mean, as specified in figure captions. For visualization only, population profiles were smoothed using a Gaussian kernel with $\sigma = 6°$.

### B.4 MODEL POPULATION RESPONSES

To compute population response profiles for the model, each latent variable $z_i$ was assigned an orientation preference. This was achieved by synthesizing a dataset of grating stimuli spanning multiple orientations and phases. For each latent unit, responses were averaged across phases to construct an orientation tuning curve.

Each tuning curve was fit with both a von Mises function and a constant function. A latent unit was classified as orientation selective if the coefficient of determination ($R^2$) of the von Mises fit exceeded that of the constant fit by at least 0.5. Using this criterion, 1415 out of 1799 latent units were classified as orientation selective. Example receptive fields and tuning curves are shown in Appendix Fig. 5.

Orientation-selective latent units were then ordered by their preferred orientation. Population response profiles were computed by binning units into $5°$ orientation bins and averaging responses within each bin. Variability across model neurons was quantified using $95\%$ confidence intervals obtained via bootstrap resampling across latent units.

### B.5 CALCULATING MOMENTS FOR PRODUCTS OF LAPLACIAN DISTRIBUTIONS

We are interested in computing the expectation values

$$\mathbb{E}[z_i], \quad \mathbb{E}[|z_i|],$$

under an unnormalized distribution of the form

$$f(z) \;\propto\; q(z; \mu, \sigma) \, \frac{p_T(z; 0, \sigma_T)}{p_0(z; 0, \mathbf{1})}, \quad z \in \mathbb{R}^N.$$

Here $q$ is a multivariate Laplace with mean vector $\mu \in \mathbb{R}^N$ and scale vector $\sigma \in \mathbb{R}^N$, $p_T$ is a zero-mean Laplace with scale vector $\sigma_T \in \mathbb{R}^N$, and $p_0$ is the standard zero-mean, unit-scale Laplace.

Because Laplace densities factorize across coordinates, both expectations reduce to one-dimensional problems of the form

$$f_i(z_i) \;\propto\; \exp\left(-\frac{|z_i - \mu_i|}{\sigma_i} - \frac{|z_i|}{\sigma_{T,i}} + |z_i|\right).$$

The integrand is piecewise exponential with kinks at $z_i = 0$ and $z_i = \mu_i$. On each segment,

$$f_i(z_i) = \exp(d + \kappa z_i),$$

with slope $\kappa$ and shift $d$ determined by the interval and parameters $(\mu_i, \sigma_i, \sigma_{T,i})$. The normalization and required moments can then be expressed in closed form.

For a segment $(z_{\mathrm{lo}}, z_{\mathrm{hi}})$ (assuming that $z_{\mathrm{lo}}$ and $z_{\mathrm{hi}}$ have the same sign, or that one of them is zero, so that the integration interval does not cross zero):

- **Normalization:**

$$N_{\mathrm{seg}} = \begin{cases} \frac{e^{d + \kappa z_{\mathrm{hi}}} - e^{d + \kappa z_{\mathrm{lo}}}}{\kappa}, & \kappa \neq 0, \\ e^d(z_{\mathrm{hi}} - z_{\mathrm{lo}}), & \kappa = 0 \end{cases}$$

- **First moment:**

$$M_{\mathrm{seg}} = \begin{cases} \frac{z_{\mathrm{hi}} e^{d + \kappa z_{\mathrm{hi}}} - z_{\mathrm{lo}} e^{d + \kappa z_{\mathrm{lo}}}}{\kappa} - \frac{e^{d + \kappa z_{\mathrm{hi}}} - e^{d + \kappa z_{\mathrm{lo}}}}{\kappa^2}, & \kappa \neq 0, \\ \frac{e^d}{2}(z_{\mathrm{hi}}^2 - z_{\mathrm{lo}}^2), & \kappa = 0 \end{cases}$$

- **Absolute-value moment:**

$$A_{\mathrm{seg}} = \begin{cases} \frac{|z_{\mathrm{hi}}| e^{d + \kappa z_{\mathrm{hi}}} - |z_{\mathrm{lo}}| e^{d + \kappa z_{\mathrm{lo}}}}{\kappa} - \frac{\mathrm{sgn}(z_{\mathrm{hi}}) e^{d + \kappa z_{\mathrm{hi}}} - \mathrm{sgn}(z_{\mathrm{lo}}) e^{d + \kappa z_{\mathrm{lo}}}}{\kappa^2}, & \kappa \neq 0, \\ \frac{e^d}{2}(z_{\mathrm{hi}}|z_{\mathrm{hi}}| - z_{\mathrm{lo}}|z_{\mathrm{lo}}|), & \kappa = 0 \end{cases}$$

By summing across all segments $(-\infty, 0)$, $(0, \mu_i)$, $(\mu_i, \infty)$ for $\mu_i > 0$ (and symmetrically otherwise), we obtain

$$N_i = \sum_{\mathrm{segments}} N_{\mathrm{seg}}, \quad \mathbb{E}[z_i] = \frac{\sum_{\mathrm{segments}} M_{\mathrm{seg}}}{N_i}, \quad \mathbb{E}[|z_i|] = \frac{\sum_{\mathrm{segments}} A_{\mathrm{seg}}}{N_i}.$$

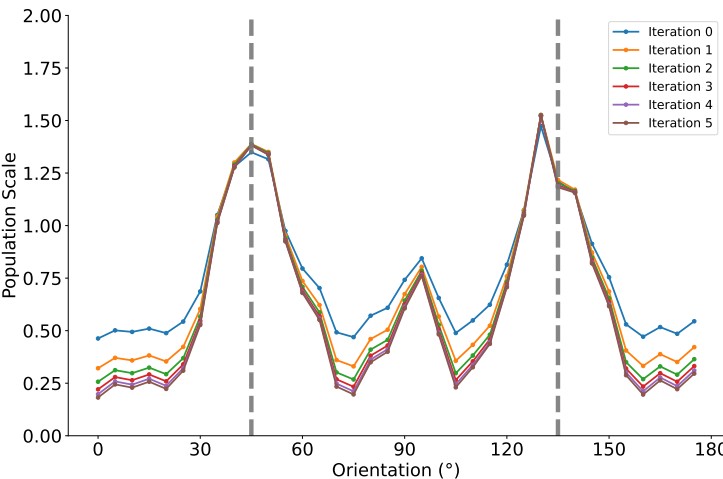

Figure 7: Convergence of the iterative solution of the self-consistency equation for the scale parameters.

To avoid numerical overflow/underflow in practice before evaluating exponentials, we subtract the maximum exponent among all segment endpoints. This does not affect the ratios, but makes the calculation stable.

### B.6 CONVERGENCE OF THE LAPLACE PRIOR FITTING

When we performed the iterative determination of the scale components (the solution of Eq. 14) we found convergence after five iterations. In Fig. 7, we plot the progression of the scale. The components of the scale vector are categorized into orientations in the same way as the model neurons.

### B.7 ALTERNATIVE PRIOR CHOICES

We investigated alternative hypotheses about the mice's internal prior across D2–D6. As shown in Appx Fig. 8b, when the model uses a 135° NoGo prior, the response profiles differ qualitatively from the measured profiles. A similar mismatch appears when we model an animal that updates its prior daily (Appx. Fig. 8a). This is consistent with the false alarm rate (the rate at which the animal licks in response to a NoGo signal) across days: on D2 the animal performs similarly to D1 (Appx. Fig. 9a), indicating adaptation to the new environment, but on subsequent days performance worsens. The animal appears to be improving throughout D2 as it adapts to the new NoGo signal (Appx. Fig. 9b).

### B.8 PEAK POSITION DEVIATION ACROSS TEST-STIMULI

To statistically quantify the distortion caused by the prior–stimulus mismatch in this model configuration, we tested whether the peaks of the response profiles were shifted away from the stimulus by the prior. To obtain model samples for this analysis, we constructed sample curves by selecting one neuron per orientation bin.

We found that the absolute difference between the peak position and the stimulus orientation was significantly larger in the TAVAE than in the naive VAE for all stimuli except 90°, where there was no prior-stimulus mismatch (Fig. 10; one-tailed t-test, $p < 10^{-4}$, $n = 40$).

### B.9 CONTRAST DEPENDENCE IN TAVAE AND ALTERNATIVE MODEL

The rule for combining the model posterior (trained on natural statistics) with the adapted prior (Eq. 6) specifies that the width of the VAE posterior determines the strength of the influence exerted by

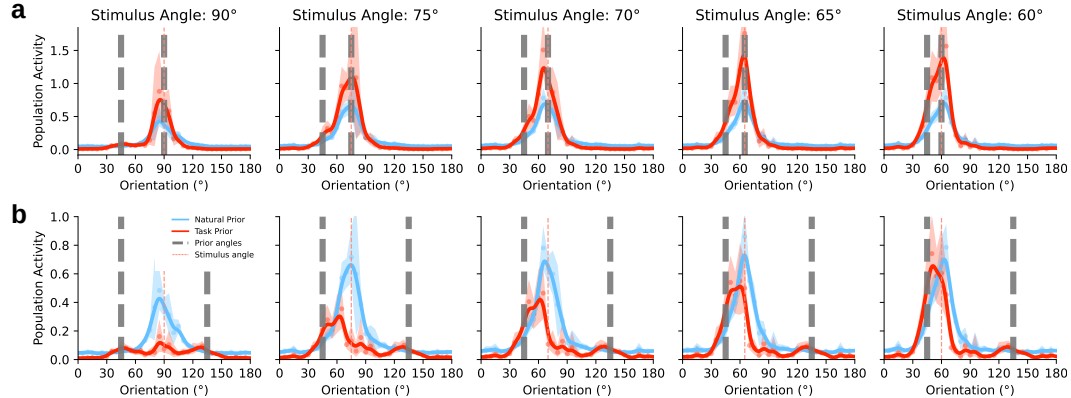

Figure 8: **a**, Population response profile of TAVAE on D2-D6 sessions for priors matched with the actual stimulus set, i.e. faithfully following the across-day changes of the NoGo stimulus. **b**, Same as **a** but for invariant prior across days, with prior reflecting the D1 conditions: Go at 45°, NoGO at 135°. *Grey dashed lines*: Prior positions; *red dashed line*: Stimulus orientation.; *Red line*: Task-trained prior condition; *Blue line*: Natural prior condition. All panels: data points, smoothing and confidence intervals as in Fig. 2.

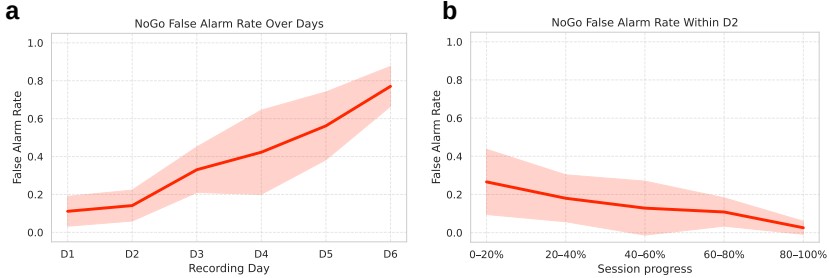

Figure 9: **a**, D2-D6 animal behaviour characterized through the false alarm rate of the NoGo stimuli. **b**, Gradual adaptation of the animals to the new NoGo stimulus of D2. False alarm rates at different quintiles of the experimental session.

the new prior. The experiment used low-contrast test gratings. In biological neurons, lower-contrast images elicit V1 responses with higher noise correlations (Orbán et al., 2016). However, in VAEs without a scaling factor, the contrast dependence of the posterior does not capture this property (Catoni et al., 2024).

We illustrated the effect of varying contrast in the TAVAE used in the paper in Fig. 11, as well as in a variant of TAVAE whose base VAE lacks a scaling factor (Fig. 12). As shown, when a scaling factor is present, the influence of adapting to a new prior increases as contrast decreases. In contrast, the model without the scaling factor exhibits only a weaker modulation by the prior, even at low contrast, compared to the TAVAE.

We quantified the model predictions in Table 2 similarly to that in Table 1 using the correlation coefficient between the model orientation bin prediction and the experimental value of that bin. We compared this metric when the model inferred its neuron responses from high- and low-contrast test images. Additionally, we also computed the correlation with a model in which the base VAE did not include the scaling factor. We can see that the correlation between the experimental response modulation and the predicted modulation is only slightly better for the TAVAE with the scale factor than for the TAVAE without the scale factor.

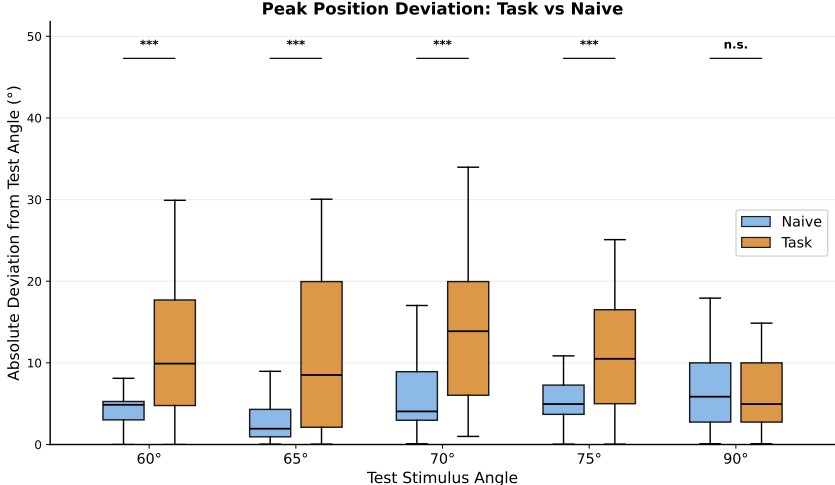

Figure 10: Absolute deviation of the peak of the population response profile from the NoGo stimulus for different NoGo stimulus angles presented on different days. Asterisks denote significant deviations in the task condition relative to the naive case.

|  | TAVAE ($c = 0.3$) | TAVAE ($c = 1.0$) | TAVAE (w/o scaling, $c = 0.3$) |
|---|---|---|---|
| $r$(TASK) | $0.78 \pm 0.02$ | $0.61 \pm 0.10$ | $0.67 \pm 0.07$ |
| $r$(TASK–NAIVE) | $0.58 \pm 0.09$ | $0.52 \pm 0.10$ | $0.54 \pm 0.09$ |

Table 2: Correlation between model and experimental population activity across orientation bins (averaged over stimuli). The standard error of the mean is shown. Correlations between experimental and model response profiles are compared for different TAVAE variants. The full TAVAE is evaluated at both reduced and full contrast levels. In addition to the full model, a constrained variant without the scaling variable is also tested. Top row: correlation between task-engaged experimental data and model predictions. Bottom row: correlation between experimental modulation (difference between task-engaged and naive conditions) and model modulation (difference between TAVAE with and without the prior).

### B.10 NEURAL PREDICTIVITY AND CKA ESTIMATES

We quantified the similarity of model predictions and experimental measurements through the correlation between model and experiment across stimuli. We established both linear Centered Kernel Alignment (CKA) (Kornblith et al., 2019) and Neural Predictivity (correlation between predicted and measured responses across held-out stimuli) Nayebi et al. (2023). The stimuli in this analysis encompassed stimuli used across D1-D6, and included the Go stimulus from D1 along with the NoGo stimuli across experimental sessions (moving gratings with orientations of 45°, 60°, 65°, 70°, 75°, 90°, and 135°). This stimulus set is considerably smaller than the sets typically used in passive viewing experiments, which is explained by the limitation of the task that animals are exposed only to a very specific set of stimuli.

As experimentally recorded neurons are solely characterized through their preferred orientation, responses of both recorded and model neurons with the same orientation preference were aggregated into 5° bins, and all metrics were calculated for averaged responses of neurons belonging to the same bin. Preferred orientations for experimentally recorded neurons were obtained from a separate block of trials. To eliminate noise-dominated responses from the analysis, bins that were far off from the task stimuli were not included in the analysis. Consequently, similar to Section 3, we used the range of bins spanning the (30°, 105°) interval.

We performed three comparisons between model and experiments. A baseline analysis was performed with the standard VAE model, using data from non-trained (naive animals). As the naive animals do not display evident biases, a standard VAE is expected to faithfully predict neuronal ac-

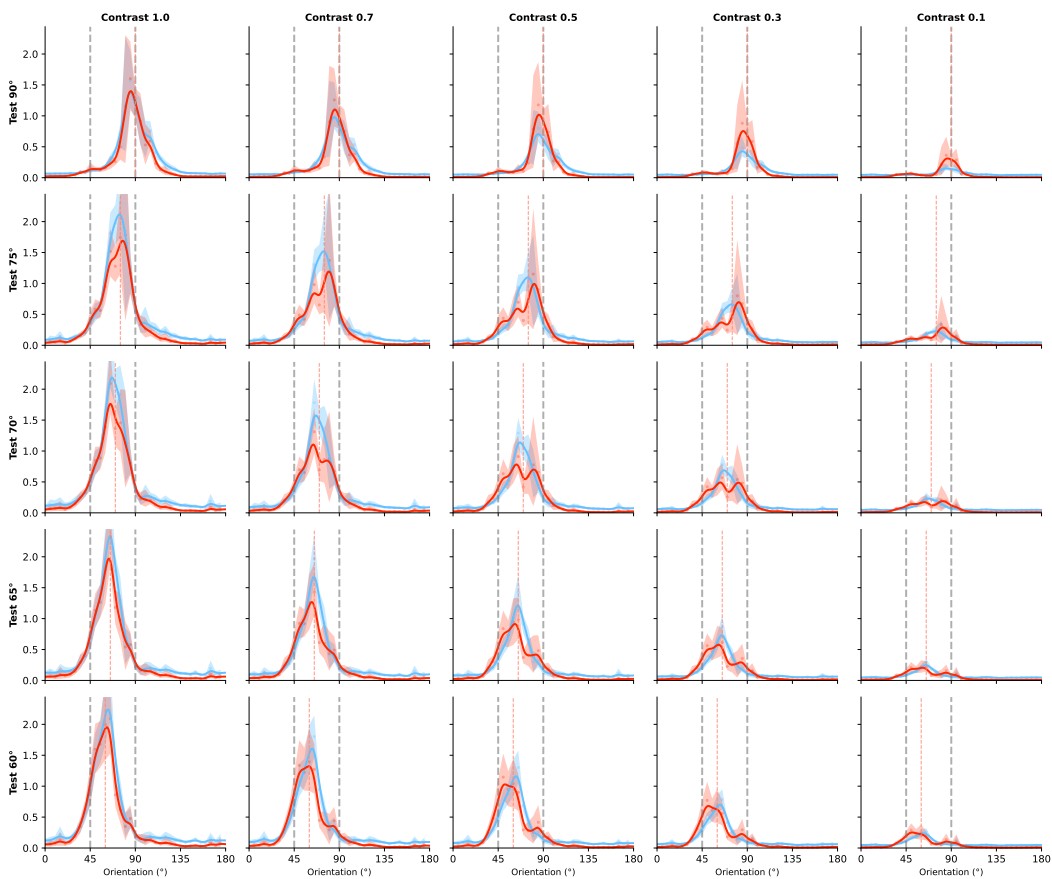

Figure 11: Contrast dependence of population responses of TAVAE for different test stimuli. *Grey dashed lines*: Prior positions; *red dashed line*: Stimulus orientation.; *Red line*: Task-trained prior condition; *Blue line*: Natural prior condition. Data points, smoothing and confidence intervals as in Fig. 2.

tivity in this condition. The standard analysis compared the TAVAE model to the neuronal responses recorded across D1-D6 in six sessions. A control was performed by using the standard VAE that cannot reflect task-specific response patterns to analyze data obtained from recordings during task execution.

CKA (Table 3) and NP (Table 4) were performed in two ways. First, a trial-averaged analysis was performed, where the experimental value was the average activity across all trials for each condition. Second, a trial-based analysis was performed, where model predictions were compared to individual experimental trials (40 trials per stimulus) and then averaged across trials. In this second scenario, we also computed a noise ceiling, comparing trial-averaged experimental responses to individual trials.

|  | VAE (on naive data) | TAVAE (on trained data) | VAE (on trained data) |
|---|---|---|---|
| CKA (Mean) | 0.85 | 0.86 | 0.82 |
| CKA (Trials) | $0.72 \pm 0.01$ (0.90) | $0.71 \pm 0.01$ (0.77) | $0.64 \pm 0.01$ (0.77) |

Table 3: Centered Kernel Alignment (CKA) scores for VAE and TAVAE models evaluated on naive and trained mouse experimental data. Top row: CKA for the mean orientation-bin activity across all available trials. Bottom row: trial-based analysis (with the standard error of the mean indicated). Values in parentheses denote the estimated noise ceiling for the corresponding dataset. The VAE prediction on the task data serves as a control.

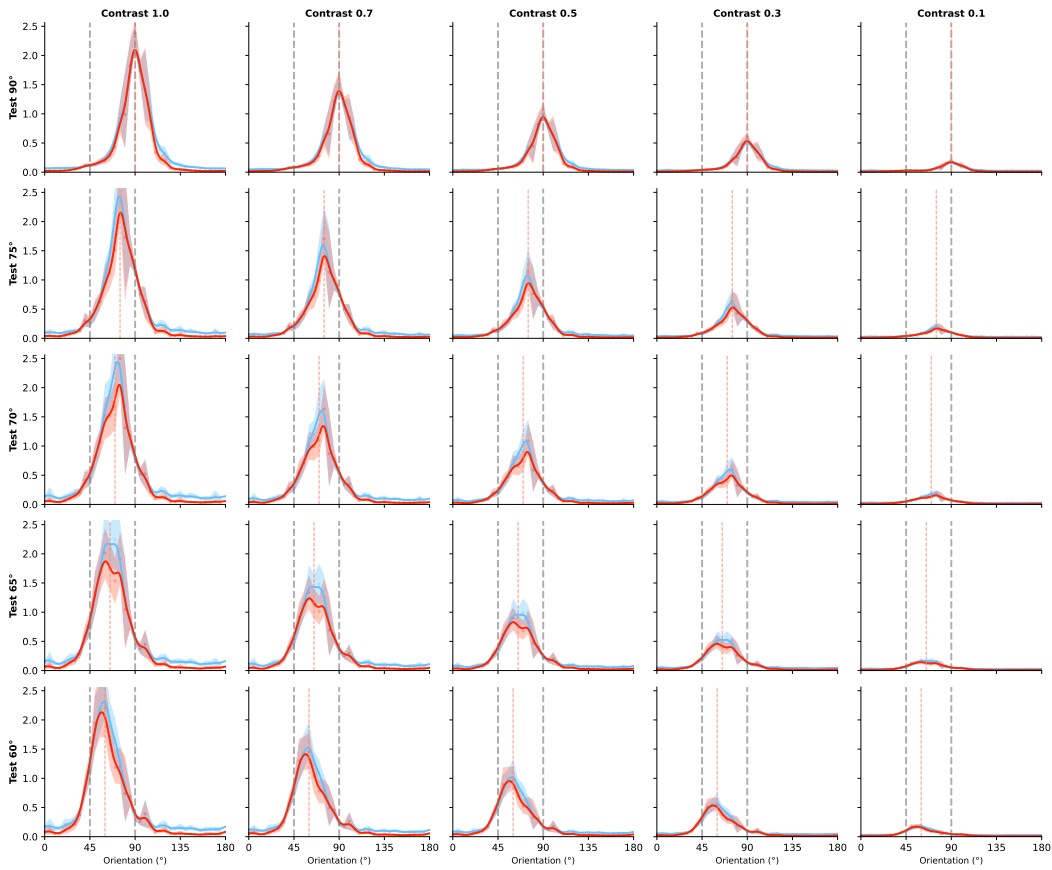

Figure 12: Contrast dependence of population responses for different test stimuli in a TAVAE lacking the scaling variable from the baseline model component. *Grey dashed lines*: Prior positions; *red dashed line*: Stimulus orientation.; *Red line*: Task-trained priors; *Blue line*: Natural prior. Data points, smoothing and confidence intervals as in Fig. 2.

|  | VAE (on naive data) | TAVAE (on trained data) | VAE (on trained data) |
|---|---|---|---|
| N.P. (Mean) | 0.907 | 0.804*** | 0.777 |
| N.P. (Trials) | 0.655 (0.772) | 0.387* (0.528) | 0.379 (0.528) |

Table 4: Neural predictivity (N.P.) scores, averaged over orientation bins, for VAE and TAVAE models evaluated on naive and trained mouse experimental data. Top row: trial-averaged analysis. Bottom row: trial-based analysis. Values in parentheses show the estimated noise ceilings of the corresponding dataset. Asterisks indicate significant differences between **TAVAE (on trained data)** and **VAE (on trained data)**, based on paired $t$-tests across orientation bins: $*, p = 0.015$ ($n = 16$); $***, p < 10^{-4}$ ($n = 16$).

CKA (Table 3) and neural predictivity (Table 4) consistently show that:

1. The VAE, when compared with experimental data from naive mice, captures meaningful variance for our grating stimuli, supporting its use as a sensible baseline for a V1 model in the passive viewing condition.

2. For data from task-engaged animals, the TAVAE better accounts for the neural responses than the VAE. However, the specificity of these metrics—primarily developed for larger stimulus sets—is lower than that of the metric used in Section 3 and Table 1.

### B.11 Estimating likelihood and prior in the space of stimulus orientations

Assuming that we observe an orientation posterior in the orientation space when recording a population of neurons, we intend to infer the likelihood and orientation prior in this space from the population response profile. Importantly, this prior–posterior formulation is distinct from that of the VAE framework, which is defined over a high-dimensional neuronal activation space rather than a one-dimensional orientation space.

We focus on D2 responses and focus on one of the modes flanking the dominant mode centered on the stimulus. The flanking mode is a mixture component of the posterior, which is a combination of the contextual prior (pr), and the likelihood (l). Approximating the posterior, likelihood, and prior with a Gaussian in the orientation space, the mean of the mode can be obtained:

$$\mu_{\text{post}} = \frac{\frac{1}{\sigma_{\text{pr}}^2}\mu_{\text{pr}} + \frac{1}{\sigma_{\text{l}}^2}\mu_{\text{l}}}{\frac{1}{\sigma_{\text{pr}}^2} + \frac{1}{\sigma_{\text{l}}^2}} = \frac{\sigma_{\text{pr}}^2\mu_{\text{l}} + \sigma_{\text{l}}^2\mu_{\text{pr}}}{\sigma_{\text{pr}}^2 + \sigma_{\text{l}}^2} \tag{16}$$

In a coordinate system centered around the NoGo stimulus of D2, some of the parameters can be readily determined: $\mu_{\text{l}} = 0$; $\mu_{\text{pr}} = 45°$; and $\mu_{\text{post}}$ can be measured from the population response profile.

We rely on the insight that if we know the means of the prior and the likelihood – which we do know – then it is the relative width of the likelihood and the prior that determines the place of the posterior. Let's assume that the prior width is $\sigma_{\text{pr}} = r \cdot \sigma_{\text{l}}$. Then Eq. 16 becomes

$$\mu_{\text{post}} = \frac{r^2\sigma_{\text{l}}^2\mu_{\text{l}} + \sigma_{\text{l}}^2\mu_{\text{pr}}}{\sigma_{\text{l}}^2(1 + r^2)} = \frac{r^2 \cdot \mu_{\text{l}} + \mu_{\text{pr}}}{1 + r^2} \tag{17}$$

Thus, if we know $\mu_{\text{post}}$, then we can calculate $r$.

We can solve for $r$ in Eq. 17 via:

$$\mu_{\text{post}} = \frac{r^2\mu_{\text{l}} + \mu_{\text{pr}}}{1 + r^2}$$
$$\mu_{\text{post}}(1 + r^2) = r^2\mu_{\text{l}} + \mu_{\text{pr}}$$
$$\mu_{\text{post}} + r^2\mu_{\text{post}} = r^2\mu_{\text{l}} + \mu_{\text{pr}}$$
$$r^2\mu_{\text{post}} - r^2\mu_{\text{l}} = \mu_{\text{pr}} - \mu_{\text{post}}$$
$$r = \sqrt{\frac{\mu_{\text{pr}} - \mu_{\text{post}}}{\mu_{\text{post}} - \mu_{\text{l}}}}$$

**Establishing the width of the likelihood.** At this point, we can establish the actual widths of the likelihood and the prior. Consider the bump in the population activity profile when presenting the Go stimulus. There the prior and the likelihood have the same mean. In such a case the posterior is a product of two Gaussians:

$$\mathcal{N}(o; \mu_{\text{Go}}, \sigma_{\text{Go}}) \propto \exp\left(-\frac{1}{2\sigma_{\text{pr}}^2}(o - \mu_{\text{Go}})^2\right)\exp\left(-\frac{1}{2\sigma_{\text{l}}^2}(o - \mu_{\text{Go}})^2\right) = \tag{18}$$

$$= \exp\left(-\frac{1}{2}\frac{1 + r^2}{r^2\sigma_{\text{l}}^2}(o - \mu_{\text{Go}})^2\right) \tag{19}$$

This highlights that the posterior width is

$$\sigma_{\text{post}} = \frac{r}{\sqrt{1 + r^2)}}\sigma_{\text{l}} \tag{20}$$

Using this, we obtain an estimate of the widths of the likelihood and prior from the width of the Go responses.

### B.12 Use of LLMs

We occasionally utilized LLMs to refine the paper's text. Additionally, we employed LLM-powered tools for programming, particularly when creating scripts for the figures.

