# OpenReview forum: "TAVAE: A VAE with Adaptable Priors Explains Contextual Modulation in the Visual Cortex"
_ICLR.cc/2026/Conference — ICLR 2026 Poster_

### Official Review · Reviewer_kx1v · 2025-10-18

**Soundness:** 3
**Presentation:** 2
**Contribution:** 3
**Rating:** 6
**Confidence:** 3

**Summary:**

The paper suggests a modified image autoencoder to account on task adaptations in visual cortex.
They have a two-stage procedure when first the latent representations are trained for natural images and then they are adjusted with respect to the task prior (which is optimized).
In this particular paper they use mice performing go/no go tasks and show that the latent space of the visual autoencoder after introducing a task prior shows same qualitative phenomena as the actual responses in V1 when the task is changing, supporting the claim that the brain performs a probabilistic inference under a prior.

**Strengths:**

1. **Elegant framework**. Beautiful idea - fixing the likelihood $p(x|z)$ and only learning a new task-specific prior $p_T(z)$ is both elegant and powerful.  The paper makes a clear hypothesis: systematic biases in V1 during task performance are the result of probabilistic inference under a learned, task-specific contextual prior. The model provides a concrete implementation of this hypothesis and generates specific, falsifiable predictions that are then confirmed by the experimental data.
2. **Qualitative comparisons**. The model reproduces the qualitative phenomena, eg splitting the distribution from unimodal to bi-modal when there is a mismatch between the train and test data (e.g. Fig 3)
3. **Reproducibility**.  The code is provided in the supplementary materials.
4. **Statistical rigor**. All the plots and tables report error-bars.

**Weaknesses:**

1. **Clarity**. The paper might benefit from a more clear high-level framework introduction, before getting to the formalism. If I get it correctly, then the autoencoder model is trained on images only and neural responses are used for validation only.
2. **Lack of quantification of qualitative results**. While Fig 3 generates nice qualitative insights, some statistical tests might support the claims, eg Hartigan's Dip Test to quantify when the red line stops being unimodal (and if it happens faster in real mice or in the model), one-Sample t-test to test that the peaks of the response are significantly shifted away from the actual stimulus orientation, and some pearson correlation to check how good the model predictions fit for the actual neuronal responses.
3. **Representations alignment is not considered**. Lines 212-214 make an assumption that the autoencoder latent space $z$ is assumed to correspond to neural activities in V1, however, this correspondence is clearly violated by the fact that $z$ could be negative. Hence, this raises questions about the validity of this assumption and how aligned the representations are in general.
4. **Limited direct applicability**. While this is a beautiful hypothesis testing framework, applying it for different stimuli can be very complicated. Specifically, eq (11) is nice and tractable as *"in a typical gratings dataset we expect a symmetry in z around zero"* (247-252). However, it is not that clear how to set up a meaningful prior in case of other tasks and out-of-distribution designs  (like distinguishing images by colors, primarily direction of "random" moving dot stimuli, etc)

Minor:
1. Inconsistent font sizes in the plots (see Fig 1 panel D and H, or Fig 2 panel A and E).

**Questions:**

1. If I get it correctly, then the autoencoder model is trained on images only and neural responses are used for validation only. Is it right? Also, you first train an autoencoder using eq (9) as the loss function to get $q(z|x)$ and $p(x|z)$ and then you only train $\underline{\sigma}_{T}$  (line 253) ?

And it adjusts $q(z|x)$ to $q_{T}(z|x)$ ? Are there any other parts retrained?

2. Why the trained baseline activity in Fig2 H is negative? I though you are taking the absolute values (lines 237-240)
3. Why exactly  does Laplace prior give us localized, oriented receptive fields? (lines 226-228)
4. Lines 237-240 identify that $z$ could be negative, which clearly misaligns the autoencoder latent space with the neuronal responses. Have you tried to restrict the $z$ to be strictly non-negative during training?
5. Connected to the previous question - lines 212-214  say that *"activations of latent variables, z, of the generative model were assumed to correspond to activations of individual neurons in V1"*. How adequate is this assumption? Have you tried to regress learn linear regression from $z$ to the actual neuronal responses (like "neural predictivity" in [1] ) and see how well it performs or do something like CKA analysis [2-4]?
6. How exactly was the neuronal data pooled across sessions? Have you selected the neurons which were enough orientation selective and then just averaged them across all sessions to make the lines in Fig 2 e, f for example? And for the autoencoder - you always used a single model (e.g. there were no several autoencoders to match the latent space for the number of neurons per session)?
7. I would appreciate your thoughts on weakness 4.

Minor:
1. What are blue and green lines in Fig1 h?

References:
[1] Nayebi, Aran, et al. "Mouse visual cortex as a limited resource system that self-learns an ecologically-general representation." PLOS Computational Biology 19.10 (2023): e1011506.
[2] Murphy, Alex, Joel Zylberberg, and Alona Fyshe. "Correcting biased centered kernel alignment measures in biological and artificial neural networks." arXiv preprint arXiv:2405.01012 (2024).
[3] Williams, Alex H., et al. "Generalized shape metrics on neural representations." Advances in neural information processing systems 34 (2021): 4738-4750.
[4] Chun, Chanwoo, et al. "Estimating Neural Representation Alignment from Sparsely Sampled Inputs and Features." arXiv preprint arXiv:2502.15104 (2025).

---

> ### Author Response · Authors · 2025-11-22
> **Addressing Weakness 1**
>
> We thank the reviewer for their positive words and enthusiasm. We also thank you for the extensive comments to improve the manuscript. We believe that the weaknesses listed can be thoroughly addressed.
>
> ### **Weaknesses**
>
> **1. Manuscript structure and clarity:** Thank you for giving us a chance to streamline the manuscript. We have restructured the Abstract, Introduction, and Methods sections. We now integrate the machine learning component, its application to neuroscience, and the experimental setting more clearly. In the Abstract and Introduction, we put more emphasis on how the machine learning model can be tied to neural responses and explain how neural responses can be interpreted in the framework of probabilistic inference in a latent variable generative model.
>
> We clarify the reviewer’s question about the way the model was optimized (see also below). We added a preamble to the Theory section that provides a broader overview. This is followed by the original section on the theory of TAVAE. In the updated manuscript, the general theory is followed by the description of how the variational autoencoder is applied to modeling V1. Finally, we present a streamlined narrative in which we present the task as in the experiment and for TAVAE side by side.
>
> With respect to training, the reviewer is correct that we follow a completely normative approach for modeling: we first train a baseline generative model (VAE) on natural image patches by optimizing the ELBO. Next, we model the adaptation of the prior to the task using synthetic grating images. Finally, we compare the prediction (coming from the new task posterior) with the experimental data.
>
>
> With respect to training, indeed, the reviewer is right, that we follow a completely normative approach for modelling:
> We first train a baseline generative model (VAE) on natural image patches optimizing ELBO. Next, we model the adaptation of the prior to the task using synthetic gratings images. Then compare prediction (coming from the new task posterior) with the experimental data.

---

> ### Author Response · Authors · 2025-11-22
> **Addressing Weakness 2**
>
> We fully agree with the reviewer that adding statistical tests for **Fig. 3** and **Fig. 4** is critical for strengthening our argument. Accordingly, we have included the following analyses:
>
> **Peak position shift test:** We show that the distance between the peak position and the stimulus position is significantly greater in the task context than in the naïve context. For the model, individual sample configurations were generated by selecting one model neuron from each orientation bin and determining the peak position of the resulting sample curve (see **Fig. 3a**, **Fig. 10**, main text under “Systematic biases of the contextual priors for OOD stimulus”).
>
> **Bimodality (local minimum) test:** To quantify the presence of bimodality with a local minimum at the stimulus, we evaluated whether the population activity exhibited a local minimum in the vicinity of the stimulus position. We identified the positions of maximal mean activity on the left and right sides of the stimulus and compared the corresponding sample values to the activity at the stimulus position. For the 70° stimulus, the flank maxima exceeded the activity at the stimulus position on both sides, indicating the presence of a local minimum between the two flanking directions. The samples were drawn from individual model neuron responses within the orientation bins used for the comparison (see **Fig. 3ab**, main text under “Systematic biases of the contextual priors for OOD stimulus”).
>
> **Shoulder asymmetry test (for the findings in Fig. 4):** We tested whether the difference in activity between the 45° and 135° orientation bins was significantly larger at Q5 (experiment) and with γ = 0.75 (model) than at Q1 or γ = 0.25 (main text under “Signature of updating the contextual prior”).
>
> **Pearson correlation:** In addition to studying the characteristic distortions of the population response curve caused by contextual modulation, we performed an additional analysis assessing the Pearson \( r \) correlation between the model and experimental mean responses (aggregated across orientation bins). Specifically, we computed this correlation for the task-engaged condition (using the TAVAE as the model prediction) and for the contextual modulation itself—defined as the correlation between the task-engaged minus naïve difference in the experimental data and the corresponding model response from the TAVAE, with and without the prior switch. For the modulation, TAVAE (with the correct prior assumption) showed significant correlation with the measurements. Also, for the task responses themselves, TAVAE showed higher correlation than the naive VAE. See **Table 1** and main text (“Systematic biases of the contextual priors for OOD stimulus” in the Results and “Characterization of responses in mice and model” in the Theory section).

---

> ### Author Response · Authors · 2025-11-22
> **Addressing Weakness 3 and 4**
>
> **3. Real-valued latent variable model:** We thank the reviewer for raising this point and giving us an opportunity to clarify, which we now do in the text (lines 217–222). We also recite the argument here. Latent variable generative models have extensively been used for modeling perceptual processes in sensory cortices and in the visual cortex in particular. Apart from positive-valued latent variable models (such as a Poisson-VAE), studies argued that a viable approach is to assume that latent variable activations correspond to the membrane potential of individual neurons.
>
> Apart from the benefit of yielding real-valued normally distributed stochastic variables, a stronger motivation is that many results obtained for spiking responses can be identified in membrane potentials too (Orbán et al., 2016, Neuron). Note that—in contrast with a Poisson-latent generative model—this approach permits spiking statistics without the restriction of tied mean and variance. In summary, we argue that a real-valued latent variable model provides a viable approach to address neural responses. We use an absolute value mapping to obtain positive responses, but a ReLU alternative could also be explored.
>
> **4. Generalization of the formalism:** This is a very interesting point. We argue that the basic idea generalizes. Specifically, the formalism indicates that for any given generative model, the prior can be adapted to novel contexts based on **Eq. 7**. The prior for a specific task can be established by assessing posteriors associated with data from the task. The exact form can be established through the basic relationship between prior and posterior:
>
> $$
> p_T(z) = \int dx \ p_T(z \mid x) \, p_T(x),
> $$
>
> where $p_T(x)$ stands for the statistics of task stimuli, sampling which yields a Monte Carlo integral. A reasonable initial approximation can be obtained by using the variational posterior $q(z \mid x)$ from the naïve model as an initial guess for $p_T(z \mid x)$.
>
> However, we agree with the reviewer that identifying a simple form for the new task prior may require additional insight, particularly when using different types of representations, e.g., a compact, highly disentangled representation instead of a high-dimensional, complete sparse one. The overall procedure remains the same: to maximize the loss defined at the end of **App. A.1** (in the revised version).
>
> ---
>
> **Reference**
>
> Orbán, G., Berkes, P., Fiser, J., & Lengyel, M. (2016). Neural variability and sampling-based probabilistic representations in the visual cortex. Neuron, 92(2), 530–543.

---

> ### Author Response · Authors · 2025-11-22
> **Addressing Questions**
>
> ### **Questions**
>
> 1. Yes, that is correct. Both the base VAE and the task variances are trained on synthetic data using the principled loss functions (ELBO). No other parts are retrained. In principle, for new task statistics, one would need both a new prior and a new posterior. However, the formula we developed provides the new posterior once the new prior is obtained. The text has been clarified to emphasize this point (lines 066–070).
>
> 2. Panel H corresponds to experimental data (where the activity can be negative since it is measured relative to a baseline activity). Panel D corresponds to the model, where the values are positive.
>
> 3. Both earlier non-deep-generative model–based accounts (Olshausen & Field, 1996, Nature) and later VAE variants (Csikor et al., 2022, arXiv; Geadah et al., 2024, Neural Computation) highlighted that localized filters (Gabor-like receptive fields) benefit from a Laplace prior, as it translates into an L1 sparsity penalty on the latent components.
>
> 4. See above at **Weakness 3**.
>
> 5. This would be an attractive approach, but our analyses are severely limited by the limited variety of images used in the experiments. As the behavioral paradigm requires that the animals are adapted to a very constrained set of images, the image set is highly controlled, and in any given session a set of ~8 images is shown, precluding detailed mapping between individual biological and model neurons. Therefore, we compared the model and experimental predictions by averaging neurons with similar preferred orientation.
>
> 6. Yes, for the experimental curves we aggregated all trials corresponding to a given day and stimulus. Neurons with the same preferred orientation were grouped, and their responses were averaged. Similarly, in the model, we determined the preferred orientation of model neurons using synthetic gratings and binned them accordingly. Only a single model was used in this analysis.
>
> 7. See **Weakness 4** above.
>
> **Minor 1.** In **Fig. 1h**, the model response curves are shown for naive (blue) and trained (green). This is clarified in the caption of the figure.
>
> ---
>
> **References**
>
> - Olshausen, B. A., & Field, D. J. (1996). Emergence of simple-cell receptive field properties by learning a sparse code for natural images. Nature, 381(6583), 607–609.
> - Csikor, F., et al. (2022). Top-down inference in an early visual cortex inspired hierarchical Variational Autoencoder. arXiv preprint arXiv:2206.00436.
> - Geadah, V., et al. (2024). Sparse-Coding Variational Autoencoders. Neural Computation, 36(12), 2571–2601.

---

> > ### Comment · Reviewer_kx1v · 2025-11-23
> > **small follow up questions / discussion**
> >
> > Thanks a lot for your clear replies.
> >
> > I have a couple of follow up questions:
> > W3- Do you actually take the absolute values of latent state during training and inference or do you only take the absolute values for the post-hoc analysis ?
> >
> > Q2 - Where can I find the details of the data preprocessing in the manuscript? How exactly the baseline was computed and if its accounted for the temporal drift during the sessions
> >
> > Q5 I do not really understand why limited visual stimuli prevents you from making the suggested analysis. You can still try to regress the neuronal activity from the latents, averaging both latents and neuronal activity across directions if you want. Or am I missing something?

---

> > > ### Author Response · Authors · 2025-11-28
> > > **Further discussion on Q5**
> > >
> > > We thank the reviewer for pushing this point. We took this point seriously and reconsidered how an analysis like the ones proposed by the original question could be implemented. Still, it is instructive to address why we argued that the investigated paradigm sets limits to such an analysis.
> > >
> > > In the example neural predictability paper, the goal is to predict the responses of a biological neuron to a set of stimuli based on (multiple linear) fitting that neuron’s activity with the activity of a combination of model neurons on a different subset of stimuli. Typically the set of stimuli is in the order of hundreds, or thousands, which enables fitting. In our paradigm we only have a very limited number of stimuli, which prevents the direct application of the same idea.
> > >
> > > However, inspired by the reviewer’s comment, we performed an analysis that performs across-image predictions with the model similar to the above example. As biological neurons are characterized with their orientation preference, we also characterized model neurons with their orientation preference. As neurons are not distinguished by other characteristics, neurons that share orientation preference are aggregated, their activity averaged.
> > >
> > > With these ‘aggregate model neurons’ we made predictions for every stimulus in the paradigm (the invariant Go stimulus and the changing NoGo stimuli) and calculated the correlation of this prediction with the responses of aggregated responses of recorded neurons across the images. This across-image correlation is then averaged across the ‘aggregated neurons’ to obtain a neural predictability value (and compared to a noise ceiling estimated using data from individual trials). We construct an analogous variant for linear **CKA**.
> > >
> > > We performed these analyses (**NP** and **CKA**) for both the **TAVAE** and standard **VAE**, such that we calculate a baseline with a standard (task-independent) VAE on recordings from naive animals. The results are now summarized in two tables, and are integrated in the manuscript in **Appendix B8** *(lines 413-417, 1016-1085)*.
> > >
> > > We have benchmarked the analysis with the task-general VAE to see if it captures meaningful variance in responses to grating stimuli from naïve mice. The correlation we obtained indicated that the VAE serves as a reasonable baseline model for V1 under passive viewing conditions. Note, however, that these values are not directly comparable to benchmark scores obtained by fitting single-neuron data to ANN layers, as those analyses typically use natural images with much richer structure.
> > >
> > > Our analyses with the TAVAE, using the NP and CKA metrics, verified that the TAVAE better explains neural responses than the VAE when evaluated on data from task-engaged animals. However, the specificity of these metrics—originally developed for larger and more diverse stimulus sets—is lower than that of the metric used in the main text (per-image population correlation).

---

> ### Author Response · Authors · 2025-11-28
> **Further discussion on W3 and Q2**
>
> **W3:** The training and inference on a given image is done on the non-absolute valued z (membrane potential-like) variable. Absolute value is taken to obtain a proxy for firing rate. When we aggregate over neurons within orientation bin and grating phases we are aggregating the absolute value (mimicking the aggregation of APrE values in orientation bin and timeframes). The normative motivation behind this choice is that learning a variational posterior with latent variables constrained to positive values (e.g., via ReLU or taking the absolute value) would violate the assumptions of the Laplace prior, which allows negative values. Consequently, the average posterior could never match the prior.
>
> **Q2:** We thank the reviewer for pointing this question out. The preprocessing of calcium signals was identical to the procedure described in *Corbo et al. (2025)*. Briefly, each trial baseline is computed as the median of the preceding intertrial raw calcium signal trace. It is used to compute the dF/F, that is then deconvolved using *Deneux et al. 2016*. This spike inference algorithm pipeline does account for baseline drift to model the signal; deconvolution therefore effectively gets rid of the signal amplitude changes due to baseline drift. We have now updated the text to provide the necessary background.
>
> **References:**
> *Corbo J, Erkat OB, McClure J, Khdour H, Polack P-O. Nat Comm, 2025, 16: 41*
> *Deneux T, Kaszas A, Szalay G, Katona G, Lakner T, Grinvald A, Rózsa B, Vanzetta I. Nat Comm, 2016, 7: 12190*

---

### Official Review · Reviewer_97W7 · 2025-10-31

**Soundness:** 2
**Presentation:** 1
**Contribution:** 2
**Rating:** 2
**Confidence:** 4

**Summary:**

This paper proposes TAVAE, a task-adapted VAE framework that modifies only the latent prior (not the encoder or decoder) to account for contextual modulation effects observed in mouse V1 during a visual discrimination task. By adapting the prior learned from natural images to task-specific contingencies., the model reproduces several well-known effects: sharpening, baseline suppression, and multimodal responses under stimulus-prior mismatch. The VAE is strongly constrained: linear decoder, Laplace prior, and overcomplete latent space, mirroring classic sparse-coding models rather than deep nonlinear architectures.

**Strengths:**

1. The model is a minimal model with biologically inspired constraints—linear decoder, sparse Laplace prior, overcomplete latent space, and GSM-style gain modulation while it mirrors classic models of V1 (e.g., Olshausen & Field).

2. The model qualitatively reproduces several experimentally observed phenomena using a single mechanism (prior variance reweighting).

**Weaknesses:**

1. While the paper claims that adaptation in the prior alone is sufficient to account for several task-induced changes in neural population statistics. The lack of comparison to single neuron activity left this claim speculative

2. Figure 3a: I really cannot see "drastic" difference between red and blue curves. There needs to be a metric or something to quantify how they are different.

3. Figure 4a; The curves are visually nearly identical in shape, except for slightly lower side peaks and a slightly higher center as γ increases. If all that happens is one peak increases slightly, calling it “updating the inference toward the new context” feels like a strong claim for a weak effect.

**Questions:**

1. Is it possible to extend this model to decode neural activity? like Maheswaranathan Neuron 2023?

2. The encoder is linear, with overcomplete latent dimensions, and trained under a Laplace prior. How close is it to ICA or sparse coding rather than deep encoder following by variational sampling?

3. Would you expect the same latent prior adaptation mechanism to work in tasks involving richer stimuli or additional visual features (e.g., natural scenes/motion)? Why or why not?

---

> ### Author Response · Authors · 2025-11-22
> **Addressing Weakness 1**
>
> We thank the reviewer for the precise two-point summary. Let us highlight that evidence of multimodal responses under stimulus-prior mismatch is more on the wishlist of systems neuroscientists than a widely demonstrated effect. Evidence seems to be more widespread in fMRI (Polonsky, 2000, Nat Neuro) than in neural-level recordings, although we acknowledge that some early Logothetis papers explored bimodal responses in primate electrophysiology. The reviewer identifies a number of weaknesses which we believe can be convincingly addressed.
>
> ### **Weakness 1**
>
> _(UPDATED from original response submitted on 22 November)_ We agree that a single cell-level analysis would deliver further important insights. An analysis like that would rely on the characterization of input-output properties of the neurons. This is a standard approach for non-task-engaged animals where sensitivities of neurons can be accurately mapped with extensive stimulus sets, as also shown by the paper cited by the reviewer (*Maheswaranathan, Neuron 2023*). In contrast to the passive viewing setting discussed there, we investigate task-related effects in task-engaged animals. This imposes severe limits on the richness of stimuli that we can apply: while the data we use is based on extensive recordings, the paradigm is limited to using ~8 stimuli. This limited response characterization also limits the sort of analysis that can be meaningfully performed.
>
> Nevertheless, we adopted two analysis methods to the investigated paradigm that use across-image predictions with the model. Instead of data-driven fit between model and experiment, we established correspondence between model and recorded neurons through their measured sensitivities. As biological neurons are characterized with their orientation preference, we also characterized model neurons with their orientation preference. As neurons are not distinguished by other characteristics, neurons that share orientation preference are aggregated, their activity averaged. With these ‘aggregate model neurons’ we made predictions for every stimulus in the paradigm (the invariant Go stimulus and the changing NoGo stimuli) and calculated the correlation of this prediction with the responses of aggregated responses of recorded neurons across the images. This across-image correlation is then averaged across the ‘aggregated neurons’ to obtain a neural predictability value (and compared to a noise ceiling estimated using data from individual trials). We construct an analogous variant for Centered Kernel Alignment analysis. We performed these analyses (NP and CKA) for both the TAVAE and standard VAE, such that we calculate a baseline with a standard (task-independent) VAE on recordings from naive animals. The results are now summarized in two tables, and are integrated in the manuscript in _Appendix B8_ (lines 413-417, 1016-1085).
>
> We have benchmarked the analyses with the task-general VAE to see if it captures meaningful variance in responses to grating stimuli from naïve mice. The correlation we obtained indicated that the VAE serves as a reasonable baseline model for V1 under passive viewing conditions. Note, however, that these values are not directly comparable to benchmark scores obtained by fitting single-neuron data to ANN layers, as those analyses typically use natural images with much richer structure. Our analyses with the TAVAE, using the NP and CKA metrics, verified that the TAVAE better explains neural responses than the VAE when evaluated on data from task-engaged animals. However, the specificity of these metrics—originally developed for larger and more diverse stimulus sets—is lower than that of the metric used in the main text (per-image population correlation). Our findings for this metric are described in the paragraph “Pearson (r) correlation” in our comment below on Weakness 2, and they are further presented in _Table 1_ and in the main text of our paper, specifically in the sections _“Systematic biases of the contextual priors for OOD stimulus”_ (Results) and _“Characterization of responses in mice and model”_ (Theory).
>
> To provide further insights into the task-related changes to single-cell behavior, we investigated task-adapted and task-general behavior of single model neurons. For this, we calculated tuning curves for model neurons from TAVAE to explore the effect of the prior. Specifically, we analyzed tuning curve responses of individual neurons with and without task-induced priors. This analysis is now added as _Fig. 6_.
>
> ---
>
> **Reference**
>
> Polonsky, A., Blake, R., Braun, J., & Heeger, D. J. (2000). Neuronal activity in human primary visual cortex correlates with perception during binocular rivalry. Nature neuroscience, 3(11), 1153–1159.

---

> ### Author Response · Authors · 2025-11-22
> **Addressing Weakness 2**
>
> The differences between the untrained and trained V1 responses are important, as only the trained animals exhibit troughs in their response profiles at the actual stimulus orientation, resulting in bimodal population responses. We agree with the reviewer that adequate statistical analysis is crucial, and we have updated the figure to include these statistics. The results confirm that trained animals display bimodal response profiles, with a local minimum near the stimulus orientation, and that population responses peak for neurons whose preferred orientations differ from the presented stimulus. We also characterized the similarity between the model and experimental curve with a Pearson correlation (see below).
>
> More specifically, we performed the following analyses for the model and experimental curves shown in **Fig. 3a–b**:
>
> **Peak position shift test:** We show that the distance between the peak position and the stimulus position is significantly greater in the task context than in the naïve context. For the model, individual sample configurations were generated by selecting one model neuron from each orientation bin and determining the peak position of the resulting sample curve (see **Fig. 3a**, **Fig. 10**, main text under “Systematic biases of the contextual priors for OOD stimulus”).
>
> **Bimodality (local minimum) test:** To quantify the presence of bimodality with a local minimum at the stimulus, we evaluated whether the population activity exhibited a local minimum in the vicinity of the stimulus position. We identified the positions of maximal mean activity on the left and right sides of the stimulus and compared the corresponding sample values to the activity at the stimulus position. For the 70° stimulus, the flank maxima exceeded the activity at the stimulus position on both sides, indicating the presence of a local minimum between the two flanking directions. The samples were drawn from individual model neuron responses within the orientation bins used for the comparison (see **Fig. 3ab**, main text under “Systematic biases of the contextual priors for OOD stimulus”).
>
> **Pearson \( r \) correlation:** In addition to studying the characteristic distortions of the population response curve caused by contextual modulation, we performed an additional analysis assessing the Pearson \( r \) correlation between the model and experimental mean responses (aggregated across orientation bins). Specifically, we computed this correlation for the task-engaged condition (using the TAVAE as the model prediction) and for the contextual modulation itself—defined as the correlation between the task-engaged minus naïve difference in the experimental data and the corresponding model response from the TAVAE, with and without the prior switch. For the modulation, TAVAE (with the correct prior assumption) showed significant correlation with the measurements. Also, for the task responses themselves, TAVAE showed higher correlation than the naive VAE. See **Table 1** and main text (“Systematic biases of the contextual priors for OOD stimulus” in the Results and “Characterization of responses in mice and model” in the Theory section).

---

> ### Author Response · Authors · 2025-11-22
> **Addressing Weakness 3**
>
> We have clarified the text to make the point we want to convey more direct. **Fig. 4a** uses the same visualization as all the other figures in the paper. The prediction of task-trained probabilistic inference is shown explicitly in **Fig. 4b**. The labeling and references in the text in the original version failed to precisely refer to this panel.
>
> To spell out the prediction: under uncertainty, alternative interpretations of the presented image pop up that correspond to the priors that have been extensively trained, and if the prior for one of the prior components weakens, then the peak related to that interpretation selectively dampens. This is shown in **Fig. 4b** and confirmed in the experiment. More precisely, we tested whether the difference in activity between the 45° and 135° orientation bins was significantly larger at Q5 (experiment) and with γ = 0.75 (model) than at Q1 or γ = 0.25 (main text under “Signature of updating the contextual prior”).
>
> Note that the current visualization uses γ = 0.25 and 0.75. If the second value were lower, the flanking peaks would be visually more dominant in the figure. We would like to stress that the prediction highlights a qualitatively exciting feature: selective reweighting of distinct modes.

---

> ### Author Response · Authors · 2025-11-22
> **Addressing Questions**
>
> ### **Questions**
>
> 1. We have a limited set of stimuli to characterize the response properties of neurons. This limitation is a consequence of the behavioral paradigm: in the paradigm it is crucial to adapt to a very specific set of stimuli, and response characteristics cannot be mapped with a granularity higher than the actual stimuli used. The limited “decoding capacity” concerns stimulus orientation. Actually, one of the most striking consequences of the paper is that if one trains a decoder outside the task, responses during the task will be severely distorted by the deployed priors.
>
> 2. That is a great point. The decoder is indeed reminiscent of a standard ICA. The motivation for relying on the VAE formalism was that task-specific posteriors could be obtained in our TAVAE formalism, and the formalism itself permits using this idea in more general settings (either higher visual cortices with highly nonlinear response properties) or more general task settings.
>
> 3. Yes, we expect the emergence of task-specific priors in other tasks with richer stimuli. We believe that such a paradigm can deliver extremely exciting insights, as it is unclear how the hierarchical cortex manages to deploy these priors at specific levels of the hierarchy. For instance, in a simple discrimination task that involves stimuli represented at higher visual areas (e.g., LM), it remains to be established how a task-related prior will shape V1 and LM.

---

### Official Review · Reviewer_4QfZ · 2025-11-04

**Soundness:** 2
**Presentation:** 2
**Contribution:** 2
**Rating:** 2
**Confidence:** 4

**Summary:**

The authors present a VAE framework that explicitly describes a task-dependent prior and test it's performance along side a neural data from mouse V1. The authors present their model, describe data collection, and present qualitative similarities between the activations of latent variables in their model to the spiking activity of V1 neurons measured by calcium imaging.

**Strengths:**

The paper presents an interesting and, to my knowledge, novel accounting of neural tuning properties in the face of changing stimulus statistics using the model they present in Section 2. They present this along side an approach for learning context specific priors in the variational framework. There do appear to be some qualitative similarities between neural data and model latents but validating these results is required before claims can be made about how their model maps mechanistically onto neural representations of stimuli.

**Weaknesses:**

I think there are 2 main dimensions on which this paper falls short of acceptance 1) validation of model structure, 2) statistical rigor, 3) clarity of question.

1) The authors make claims about the qualitative properties of the latents of their model and how they match those of the real data. However, I'm not sure it's possible to attribute these features (even if they are statistically valid) to the prior structure of the model exclusively. Specifically, no ablation analysis of the model was conducted to determine which of their modeling choices was essential to their findings. For example, how critical was it that the latent responses were sparse? How important was the scaling latent? Neither of these choices were evaluated in any way and it is not clear they are germane to the properties they intend to model.

2) There is virtually no statistical analysis beyond Figure 2. Error bars and shaded regions around population tuning curves are not defined. Data points in tuning curve plots (eg. red and blue dots in Figure 2a,b) are not defined. Moreover, if these really are data points and the shaded regions are supposed to be 95% confidence intervals then I suspect their inference is over-confident.

3) It's not obvious what the authors are testing when they are examining neural activity along side latent activations. This seems to be an unreasonably course level of analysis and I would not expect a clear correspondence to exist beyond something incidental. Perhaps the authors meant to examine the posterior distribution over the stimulus? This would have real cognitive meaning in the context of a shift in prior probabilities.

**Questions:**

The authors should clarify their mechanistic claims about why their model matches the data in the ways they claim, the modeling choices, statistical inference, and all claims should be accompanied by statistical tests.

---

> ### Author Response · Authors · 2025-11-22
> **Addressing Weakness 1**
>
> We thank the reviewer for the summary and for highlighting the novelty of our approach and results. We believe that the raised concerns can be fully addressed, as we explain in detail below.
>
> ### **Weakness 1**
>
> The reviewer requests a breakdown of the contribution of the different model components. Below we provide direct explanations, which we are also integrating into the manuscript.
>
> #### **Scaling variable and the effect of contrast**
>
> The Gaussian Scale Mixture–style scaling is critical for adequate handling of uncertainty in the Variational Autoencoder (VAE). In the context of the current study, contrast-related changes in uncertainty are important.
>
> As shown by Catoni et al. (2024, arXiv, **Fig. 1**), a standard VAE cannot deliver broader posteriors when contrast decreases—that is, when uncertainty increases. The rule for combining the model posterior (trained on natural statistics) with the task-adapted prior (**Eq. 7**) indicates that the width of the VAE posterior determines the strength of the influence that the new prior exerts. Since contrast can substantially affect posterior width, it plays a critical role in exploring the specific effects of the task prior. Note that our experiments used low test stimulus contrast to probe response properties. Therefore, our study required reliable inference under changing contrast.
>
> To demonstrate the contribution of changing uncertainty, we reproduced the basic experiment with varying test stimulus contrast (now included in **Fig. 11, App. B.7**). Consistent with expectations, we find that at large test stimulus contrast, the contextual modulation of the population response is substantially weaker.
>
> To test the importance of the scaling variable, we repeated the experiment using a TAVAE built upon a VAE without a scaling factor. Although modulation was still observed, at low contrast—the regime corresponding to the experimental data—the effect was weaker, and the strength of modulation was not clearly governed by contrast (see **Fig. 12, App. B.7**).
>
> We quantified the correspondence between model and experimental responses using the Pearson correlation coefficient, following the methodology described below under **Weakness 2** (see **Table 2, App. B.7**, and additional comparisons in **Table 1**). We found that removing the scaling factor slightly reduced the correlation coefficient \( r \) compared to the model with the scaling factor, though the TAVAE without scaling retained predictive power for contextual modulation. These new analyses are presented in **App. B.7**.
>
> #### **Laplace prior**
>
> Both earlier non-deep-generative accounts (Olshausen and Field, 1996, Nature) and later VAE variants (Csikor et al., 2022, arXiv) highlighted that localized filters (Gabor-like receptive fields) only appear from a Laplace prior. The learned representation is therefore better aligned with the simple cell receptive fields of the primary visual cortex, the main target of our modeling efforts. This clarification is now included in the text (lines **205–207**).
>
> ---
>
> **References**
>
> Catoni, J., Martos, D., Csikor, F., Ferrante, E., Milone, D. H., Meszéna, B., ... & Echeveste, R. (2024). Uncertainty in latent representations of variational autoencoders optimized for visual tasks. arXiv preprint arXiv:2404.15390.
>
> Csikor, F., Martos, D., Catoni, J., Echeveste, R., & Ferrante, E. (2022). Top-down inference in an early visual cortex inspired hierarchical Variational Autoencoder. arXiv preprint arXiv:2206.00436.
>
> Olshausen, B. A., & Field, D. J. (1996). Emergence of simple-cell receptive field properties by learning a sparse code for natural images. Nature, 381(6583), 607–609.

---

> ### Author Response · Authors · 2025-11-22
> **Addressing Weakness 2**
>
> ### **Weakness 2**
>
> We agree with the reviewer that thorough statistical analysis is required to underpin the claims. We have taken this comment on board and systematically reviewed our figures and performed the necessary analyses. Specifically, we have also performed the following new statistical tests (on the experimental and model data) regarding the claims made about **Fig. 3** and **Fig. 4**.
>
> **Peak position shift test:** We show that the distance between the peak position and the stimulus position is significantly greater in the task context than in the naïve context. For the model, individual sample configurations were generated by selecting one model neuron from each orientation bin and determining the peak position of the resulting sample curve (see **Fig. 3a**, **Fig. 10**, main text under “Systematic biases of the contextual priors for OOD stimulus”).
>
> **Bimodality (local minimum) test:** To quantify the presence of bimodality with a local minimum at the stimulus, we evaluated whether the population activity exhibited a local minimum in the vicinity of the stimulus position. We identified the positions of maximal mean activity on the left and right sides of the stimulus and compared the corresponding sample values to the activity at the stimulus position. For the 70° stimulus, the flank maxima exceeded the activity at the stimulus position on both sides, indicating the presence of a local minimum between the two flanking directions. The samples were drawn from individual model neuron responses within the orientation bins used for the comparison (see **Fig. 3ab**, main text under “Systematic biases of the contextual priors for OOD stimulus”).
>
> **Pearson  \( r \) correlation**: In addition to studying the characteristic distortions of the population response curve caused by contextual modulation, we performed an additional analysis assessing the Pearson  \( r \) correlation between the model and experimental mean responses (aggregated across orientation bins). Specifically, we computed this correlation for the task-engaged condition (using the TAVAE as the model prediction) and for the contextual modulation itself—defined as the correlation between the task-engaged minus naïve difference in the experimental data and the corresponding model response from the TAVAE, with and without the prior switch. For the modulation, TAVAE (with the correct prior assumption) showed significant correlation with the measurements. Also, for the task responses themselves, TAVAE showed higher correlation than the naive VAE. See **Table 1** and main text (“Systematic biases of the contextual priors for OOD stimulus” in the Results and “Characterization of responses in mice and model” in the Theory section).
>
> **Shoulder asymmetry test (for the findings in Fig. 4):** We tested whether the difference in activity between the 45° and 135° orientation bins was significantly larger at Q5 (experiment) and with γ = 0.75 (model) than at Q1 or γ = 0.25 (main text under “Signature of updating the contextual prior”).
>
> **Error bars around response curves (shaded areas):** We refined the estimation of uncertainty in the mean responses by computing 95% confidence intervals for each orientation bin using bootstrap resampling of individual model neurons within that bin. These intervals are shown as shaded bands around the curves (see **Fig. 2a–b**, **Fig. 3b**, and **Fig. 4a–b**). We note that the curves in Figs. 2 and 3 represent population response profiles characterizing the responses to specific stimuli. Individual tuning curve modulations for the model are now included as **Fig. 6** (in **App. B.1**).

---

> ### Author Response · Authors · 2025-11-22
> **Addressing Weakness 3**
>
> ### **Weakness 3**
>
> The reviewer requires us to clarify the correspondence between latent variables of the model and raises the question of whether we “are examining the posterior distribution over the stimulus.” We interpret the question as asking if the responses of biological neurons correspond to the posterior distribution emerging upon stimulus presentation. We can confirm this: upon stimulus presentation, a posterior over the latent population is calculated, and one particular latent is assumed to correspond to a model V1 neuron. This correspondence is encouraged by the matching sensitivities of biological and model neurons: when performing white noise analysis to obtain the linear response kernel, similar Gabor-like receptive fields are obtained (**Fig. 5**).
>
> Classical approaches identify neuronal responses with a Maximum a Posteriori (MAP) estimate (e.g., Olshausen & Field, 1996, Nature; Schwartz & Simoncelli, 2001, Nat Neurosci) or identify neuronal membrane potential responses with samples from this posterior (Orbán et al., 2016, Neuron; Festa et al., 2021, Nat Communications; Echeveste et al., 2020). In this paper, we do not directly commit to either interpretation, as we are analyzing long response windows, which even in a sampling interpretation would encompass multiple samples; therefore, MAP and sampling accounts make similar predictions. Nevertheless, the multimodal response profile we observe is naturally aligned with a sampling interpretation.
>
> We agree with the reviewer that interpreting V1 neuronal responses as representing a posterior has deep implications for cognitive science. We believe that signatures of a prior at the level of the primary visual cortex, which can flexibly adapt to task demands, is a particularly exciting perspective, as it complements ideas on acquiring priors through lifetime exposure to natural stimulus statistics (e.g., Berkes et al., 2011, Science). We have updated the text at multiple points to integrate this argument and increase clarity (lines **018–025**, **046–052**, **071–085**, **159–197**).
>
> ---
>
> **References**
>
> Berkes, P., Orbán, G., Lengyel, M., & Fiser, J. (2011). Spontaneous cortical activity reveals hallmarks of an optimal internal model of the environment. Science, 331(6013), 83–87.
>
> Echeveste, R., Aitchison, L., Hennequin, G., & Lengyel, M. (2020). Cortical-like dynamics in recurrent circuits optimized for sampling-based probabilistic inference. Nature neuroscience, 23(9), 1138–1149.
>
> Festa, D., Aschner, A., Davila, A., Kohn, A., & Coen-Cagli, R. (2021). Neuronal variability reflects probabilistic inference tuned to natural image statistics. Nature communications, 12(1), 3635.
>
> Olshausen, B. A., & Field, D. J. (1996). Emergence of simple-cell receptive field properties by learning a sparse code for natural images. Nature, 381(6583), 607–609.
>
> Orbán, G., Berkes, P., Fiser, J., & Lengyel, M. (2016). Neural variability and sampling-based probabilistic representations in the visual cortex. Neuron, 92(2), 530–543.
>
> Schwartz, O., & Simoncelli, E. P. (2001). Natural signal statistics and sensory gain control. Nature neuroscience, 4(8), 819–825.

---

> ### Author Response · Authors · 2025-11-22
> **Addressing Questions**
>
> ### **Questions**
>
> Thank you for raising this question. We propose to provide mechanistic insight into what happens when population response profiles are reshaped through the reshaping of response properties of individual model neurons. We now provide an additional figure that shows the response properties of individual model neurons in a trained TAVAE model (**Fig. 6**). Tuning curves of neurons tuned to different orientations show differential modulation depending on how their preferred orientation relates to trained stimulus orientations.
>
> We argue that the task-specific prior affects different neurons differently: those whose preferred orientation (peak orientation-dependent response) matches the task prior orientation display high-gain tuning curves, whereas neurons whose preferred orientation does not match the prior orientation are biased toward dampened response intensities and shallower tuning curves. This basic insight explains the shifts in the population response profile. For instance, for a stimulus that does not match either of the task stimuli, neurons with preferred orientation at the actual stimulus will have dampened responses because they are suppressed by the prior, while those neurons whose preferred orientation matches the prior will have dampened responses because the actual stimulus is at the tail of their tuning curves. Consequently, neurons with preferred orientations in between the prior and the actual stimulus can trade off the dampening from their tuning curves with the stimulus being away from the tuning curve peak.
>
> The three main effects in the population response profile we present in the paper are consequences of this insight:
> 1. Baseline suppression comes from tighter prior-related suppression of residual activity.
> 2. Bimodal responses arise from both priors contributing to the interpretation of the actual stimulus, yielding competing but biased interpretations.
> 3. Within-day transformation of the response profile results from shifting prior.
>
> We now clarify this argument in the manuscript (lines **297–332**, **316–318**).

---

> ### Comment · Reviewer_4QfZ · 2025-11-23
> **Revision of scores following rebuttal**
>
> The authors have adequately addressed my concerns and I have updated my score according to my assessment of how their updates would reflect the final quality of the paper. Thank you.

---

> > ### Author Response · Authors · 2025-11-28
> > **Thank you for the feedback**
> >
> > We thank the reviewer for the feedback and we are glad that all the concerns of this reviewer has been addressed.

---

### Author Response · Authors · 2025-12-03
**Summary of Rebuttal**

First, we would like to thank the reviewers for their thoughtful, constructive feedback and for their enthusiastic engagement with the paper.

In their initial reviews, although the reviewers appreciated the contribution of the paper (claiming it is novel and elegant, as well as appreciating that the model synthesizes a range of neuroscience phenomena), they identified weaknesses. We took these criticisms on board and addressed every single raised issue. We were fortunate to have feedback from two of the reviewers: R1 (4QfZ) confirmed that all of their concerns were addressed (raising their score to 6); R3 (kx1v) also positively received our detailed responses and came forward with follow-up clarification questions, which we answered in detail in a second round. This discussion with R3 has not concluded until shutdown but the discussion indicated that all the issues raised by them have been addressed. R2 (97W7) raised issues that concerned additional analysis and statistical corroboration of the paper’s claims. We thoroughly addressed all of these issues.

As feedback on our responses was not received from Reviewer 97W7 before the shutdown, we briefly elaborate on this review. We are confident that weaknesses #2 and #3 have been fully addressed as these are firmly aligned with issues raised by the other reviewers who confirmed that our follow-up analysis corroborated the results. Weakness #1 of Reviewer 97W7 remains an issue which we did not have a chance to discuss. With respect to this, we conducted two additional analyses standard in neuroscience pipelines but adapted to the paradigm at hand. We hope that these analyses provide convincing support that rich and striking biases in the primary visual cortex can be parsimoniously interpreted as learned task-specific priors in a generative model of a visual task.

Here we provide a summary of the updates to the manuscript:
- Restructured the Introduction and Theory sections according to the request of Reviewer kx1v, clarified the relationship between the VAE architecture and neural recordings (as required by 4QfZ), and to improve on the Presentation.
- Made sure that all claims regarding both modeling results and experimental data are supported by statistical tests (as per Reviewers 4QfZ and 97W7).
- Introduced a quantitative measure to assess the level of match between the model and experimental data, which addresses a point raised by Reviewer 4QfZ. We capitalized on this metric to quantitatively evaluate the developed TAVAE model against alternatives that ablated different components of the main model, addressing Reviewer 4QfZ’s issue.
- Added two additional quantitative analyses (CKA and neuronal predictivity) that are standard in the literature to evaluate the alignment of model predictions and experimental data, which addresses point raised by Reviewers 97W7 and kx1v.
- Provided a mechanistic explanation for the observed effects in the model, in response to Reviewer 97W7 and 4QfZ.
- To wrap up, the update resulted in 4 new tables (1 in main text, Tables 1-4), 4 new figures in the appendix (Figs 6, 10, 11, 12), 3 new sections in the appendix (B6, B7, B8), and updates to figure panels in the main text.

Finally, we have now uploaded the final revision, which includes minor clarifications and aesthetic changes.

---

### Meta-Review · Area_Chair_CM8M · 2026-01-16

**Summary:**

This paper introduces a new VAE framework for modeling neural activity responses in V1 with a task specific context. The approach is used to model V1 neural responses in mice change during a visual discrimination task. The key contribution is showing that task-specific priors can explain neural phenomena including response sharpening, baseline suppression, and bimodal responses when stimuli violate learned task statistics.

The paper initially received two strong rejects and one weak accept with initial concerns regarding statistical rigor, lack of ablations, correspondence between model latents and neural activity. The author rebuttal addressed these concerns with new statistical analyses, ablations and clarifications, leading one reviewer to increase their score to a weak accept. I suspect the third reviewer might have increased their score modestly, given the strength of the rebuttal. However, this paper still remains borderline accept.

**Reviewer Concerns:**

The paper initially received two strong rejects and one weak accept with initial concerns regarding statistical rigor, lack of ablations, correspondence between model latents and neural activity. The author rebuttal addressed these concerns with new statistical analyses, ablations and clarifications. A potentially unresolved concern has to do with the size of the effects shown, questioning how big an advance this paper constitutes.

**Reviewer Scores:**

One reviewer already recommended a weak accept and might only have increased their score modestly. The rebuttal led a second reviewer to increase their score to a weak accept. I suspect the third reviewer might have also increased their score modestly, given the strength of the rebuttal. However, this paper still remains borderline accept.

---

### Decision · Program_Chairs · 2026-01-26

Accept (Poster)